

# Quasi-hydrostatic equations for climate models and the study on linear instability

Robert Nigmatulin[1,2] and Xiulin Xu[1]

[1]Lomonosov Moscow State University, Moscow, 119991 Russia
[2]Shirshov Institute of Oceanology, Russian Academy of Sciences, Moscow, 117218 Russia

**Correspondence:** Xiulin Xu (xiulin.xu@mech.math.msu.su; xiulin.xu@icloud.com)

**Abstract.** An advanced "quasi-hydrostatic approximation" of 3-dimensional atmospheric-dynamics equations is proposed and justified with the practical goal to optimize atmospheric modelling at scales ranging from meso meteorology to global climate. For the vertically quasi-hydrostatic flow with inertial forces negligibly small compared to gravity forces, the asymptotically exact equation for vertical velocity is obtained. In the closed system of hydro/thermodynamic equations, the pressure is deter-

5 mined by the total air mass above, so that mass instead of pressure is considered as a dependent variable. In such a system, the sound waves are filtered, though the horizontal inertia forces are taken into account in the horizontal momentum conservation equations. The major practical result is an asymptotically exact equation for vertical velocity in the quasi-hydrostatic system of the atmospheric dynamics equations.

 Investigation of the stability of solutions to the system in response to small shortwave perturbations has shown that solutions

10 have the property of shortwave instability. There are situations when the growth rate of the perturbation amplitude tends to infinity. It means that the Cauchy problem for such equations may be ill-posed. Its formulation can become conditionally correct if solutions are sought in a limited class of sufficiently smooth functions whose Fourier harmonics tend to zero reasonably quickly when the wavelengths of the perturbations approach zero. Thus, the numerical scheme for the quasi-hydrostatic equations using the finite-difference method requires an adequately selected pseudo-viscosity to eliminate the instability caused by

15 perturbations with wavelengths of the order of the grid size. The result is useful for choosing appropriate vertical and horizontal grid sizes for modelling to avoid shortwave instability associated with the property of the system of equations. Implementation of pseudo-viscosities helps to smoothen or suppress the perturbations that occur during modelling.

## 1 Introduction

The advantage of quasi-hydrostatic equations is the efficiency of numerical calculations of atmospheric circulation by filtering

20 the sound wave. However, as discussed in Orlanski (1981), the aspect ratio between vertical and horizontal scales significantly influences the dispersion relation that, in turn, determines the stability of numerical calculation. Most global climate models are based on a system of dynamic equations in quasi-hydrostatic approximation White and Bromley (1995); White et al. (2005).

 The atmospheric predictability problem has been discovered and studied for a long time since the early works like (Lorenz, 1969a, 1982). Lorenz Edward proposed three approaches to this problem (Lorenz, 1969b), within which the dynamical ap-





proach is adopted in the analysis of present paper.The influence of scales on the stability (or predictability) is discussed in more recent works (Hohenegger and Schar, 2007; Tribbia and Baumhefner, 2004).

We introduce a system of 3-dimensional atmospheric-dynamics equations with a quasi-hydrostatic approximation for climate modelling. Based on this system, we investigate the linear instability in response to shortwave perturbations. The present paper focuses on this problem with the practical goal to assure realistic numerical modelling, with emphasis on the scale effects.

In this section, the primitive equations are introduced. Section 2 presents derivation of the closed quasi-hydrostatic equations with accurate approximation of vertical velocity estimation and the asymptotically exact system to the equations in quasi-hydrostatic approximation. In Section 3, the system of equations for small perturbations in dimensionless form and the cubic characteristic equation with complex coefficients are derived. In section 4, the stability of solutions to the system of equations with response to shortwave perturbations is analyzed through solving the cubic characteristic equation. Section 5 focuses on

the elimination of instability caused by the shortwave perturbations with the aid of pseudo-viscosities.

Almost the entire mass of the atmosphere is located in the layer with thickness, $H$, of order 10 km. So, the atmospheric dynamics outside polar zones can be considered in quasi-Cartesian coordinate system $x$, $y$, $z$, $t$, where $x$ and $y$ are directed horizontally along latitude and longitude, respectively, $z$ is the height over the surface, and $t$ is time.

The equations of conservation of mass and momentum read (Kochin et al., 1966; Marchuk, 1974; Brekhovskikh and Gon-

charov, 1982; Batchelor and Batchelor, 2000; Gill, 2016; Loitsyanskii, 1970; Sedov, 1997; Holton and Hakim, 2012; Nigmatulin, 2014):

$$\frac{\partial \rho}{\partial t} + \operatorname{div}(\rho \mathbf{v}) = 0, \tag{1.1}$$

$$\frac{\mathrm{d}v_x}{\mathrm{d}t} = -\frac{1}{\rho}\frac{\partial p}{\partial x} + \frac{1}{\rho}\left(\frac{\partial \Pi_{xx}}{\partial x} + \frac{\partial \Pi_{xy}}{\partial y} + \frac{\partial \Pi_{xz}}{\partial z}\right) + f_x, \tag{1.2}$$


$$\frac{\mathrm{d}v_y}{\mathrm{d}t} = -\frac{1}{\rho}\frac{\partial p}{\partial y} + \frac{1}{\rho}\left(\frac{\partial \Pi_{yx}}{\partial x} + \frac{\partial \Pi_{yy}}{\partial y} + \frac{\partial \Pi_{yz}}{\partial z}\right) + f_y, \tag{1.3}$$

$$\frac{\mathrm{d}v_z}{\mathrm{d}t} = -\frac{1}{\rho}\frac{\partial p}{\partial z} + \frac{1}{\rho}\left(\frac{\partial \Pi_{zx}}{\partial x} + \frac{\partial \Pi_{zy}}{\partial y} + \frac{\partial \Pi_{zz}}{\partial z}\right) + f_z - g, \tag{1.4}$$

$$\left(\frac{\mathrm{d}}{\mathrm{d}t} = \frac{\partial}{\partial t} + \mathbf{v}\cdot\nabla\right)$$

where $\rho$ and $\mathbf{v} = (v_x, v_y, v_z)$ are density and velocity of air, $p$ is pressure, $\Pi_{ij}$ is the viscous (turbulent) stress tensor, $\mathbf{f} =$

$(f_x, f_y, f_z)$ is the Coriolis force vector, and $g$ is the gravity acceleration ($g \approx 9.81\mathrm{m\ s^{-2}}$).

We consider air as an ideal gas, from whose state equation for the pressure $p$ and the specific internal energy $u$ are determined by the specific gas constant $R \approx 286\ \mathrm{m^2\ s^{-2}K^{-1}}$ and the adiabatic index $\gamma \approx 1.4$:

$$p = R\rho T, \quad u = c_v T + \mathrm{const}, \quad c_v = R/(\gamma - 1), \tag{1.5}$$





where $T$ is the air temperature, and $c_v$ is the specific heat capacity at constant volume.

The equation of conservation of energy in the form of the thermodynamic equation reads:

$$\frac{\mathrm{d}T}{\mathrm{d}t} = \frac{Q^*}{\rho c_{\mathrm{v}}} + \frac{p}{\rho^2 c_{\mathrm{v}}} \frac{\mathrm{d}\rho}{\mathrm{d}t}$$

$$\left( Q^* \equiv -\frac{\partial q}{\partial z} + Q - J^{(\mathrm{e})} l^{(\mathrm{e})} - J^{(\mathrm{m})} l^{(\mathrm{m})} \right),$$

(1.6)

Here $Q^*$ is total heat influx into unit volume due to the turbulent heat flux, $q$, radiation heat absorption, $Q$, and latent heat consisted of evaporation (condensation) heat of droplets, $J^{(\mathrm{e})} l^{(\mathrm{e})}$, and the heat due to melting (solidification) of ice particles, $J^{(\mathrm{m})} l^{(\mathrm{m})}$. Here, it is taken into account that the horizontal turbulent heat transfer is much smaller than the vertical one, so that

the horizontal heat transfer is due to advection, determined by horizontal velocities $v_x$ and $v_y$.

Using the state equation(1.5), the thermodynamic energy equation (1.6) can be rewritten in terms of the following relation between the time derivatives of pressure and density:

$$\frac{1}{\rho} \frac{\mathrm{d}\rho}{\mathrm{d}t} = \frac{1}{\gamma p} \frac{\mathrm{d}p}{\mathrm{d}t} - \frac{\gamma - 1}{\gamma p} Q^*.$$

(1.7)

The equations (1.1) – (1.7) form a closed system. However, for filtering the sound wave, the quasi-hydrostatic equation is

used instead of the vertical momentum equation (1.4). Thus, for the new system of equations with quasi-hydrostatic approximation, it is necessary to derive an equation to estimate the vertical velocity.

## 2    Asymptotics of hydrodynamic equations with vertical quasi-hydrostatic approximation

### 2.1    Meteorological and climate scales

We consider non-extreme hydrodynamic and thermodynamic processes in the Earth's atmosphere with climatic scales, i.e.,

time scale, $\tau$, horizontal length scale, $L_x, L_y \sim L_{\mathrm{hor}}$, vertical length scale, $L_z \sim L_{\mathrm{ver}}$, horizontal velocity scale, $V_x, V_y \sim V_{\mathrm{hor}}$, and vertical velocity scale, $V_z \sim V_{\mathrm{ver}}$, with the following values:

$$\tau \geq 10^2 \mathrm{s}, \quad L_{\mathrm{hor}} \geq 10^3 \mathrm{m}, \quad L_{\mathrm{ver}} \geq 10^2 \mathrm{m},$$

$$V_{\mathrm{hor}} \sim 10^1 \mathrm{m/s}, \quad V_{\mathrm{ver}} \sim 10^0 \mathrm{m/s}.$$

(2.1)

With the use of these scales, we introduce dimensionless values of time, $\overline{t}$, horizontal coordinates, $\overline{x}, \overline{y}$, and horizontal velocities, $\overline{v}_x, \overline{v}_y$, vertical coordinate, $\overline{z}$, and vertical velocity, $\overline{v}_z$:

$$\overline{x} = \frac{x}{L_{\mathrm{hor}}}, \overline{y} = \frac{y}{L_{\mathrm{hor}}}, \quad \overline{z} = \frac{z}{L_{\mathrm{ver}}}, \quad \overline{t} = \frac{t}{\tau},$$


$$\overline{v}_x = \frac{v_x \tau}{L_{\mathrm{hor}}}, \quad \overline{v}_y = \frac{v_y \tau}{L_{\mathrm{hor}}}, \quad \overline{v}_z = \frac{v_z \tau}{L_{\mathrm{ver}}}.$$

(2.2)





As a result, the horizontal, vertical, and Coriolis accelerations, as well as inertial forces, can be evaluated as:

$$\frac{\mathrm{d}v_x}{\mathrm{d}t} = \frac{V_{\mathrm{hor}}}{\tau}\frac{\mathrm{d}\overline{v}_x}{\mathrm{d}\overline{t}} = A_{\mathrm{hor}}\frac{\mathrm{d}\overline{v}_x}{\mathrm{d}\overline{t}},$$
$$\frac{\mathrm{d}v_y}{\mathrm{d}t} = \frac{V_{\mathrm{hor}}}{\tau}\frac{\mathrm{d}\overline{v}_y}{\mathrm{d}\overline{t}} = A_{\mathrm{hor}}\frac{\mathrm{d}\overline{v}_y}{\mathrm{d}\overline{t}}, \qquad (2.3)$$
$$\frac{\mathrm{d}v_z}{\mathrm{d}t} = \frac{V_{\mathrm{ver}}}{\tau}\frac{\mathrm{d}\overline{v}_z}{\mathrm{d}\overline{t}} = A_{\mathrm{ver}}\frac{\mathrm{d}\overline{v}_z}{\mathrm{d}\overline{t}},$$

where

$$A_{\mathrm{hor}} = \frac{V_{\mathrm{hor}}}{\tau} = \frac{V_{\mathrm{hor}}^2}{L_{\mathrm{hor}}}, \qquad A_{\mathrm{ver}} = \frac{V_{\mathrm{ver}}}{\tau} = \frac{V_{\mathrm{ver}}^2}{L_{\mathrm{ver}}},$$
$$A_{\mathrm{cor}} = \frac{V_{\mathrm{hor}}}{\tau_{\mathrm{cor}}} \sim 10^{-3}\mathrm{m/s}^2, \quad \tau_{\mathrm{cor}} = \frac{1}{2\Omega} = 0,688 \times 10^4\mathrm{s}.$$

Here $\Omega \approx 0,727 \times 10^{-4}\mathrm{s}^{-1}$ is Earth's angular velocity of rotation. Here $A_{\mathrm{hor}}$, $A_{\mathrm{ver}}$, and $A_{\mathrm{cor}}$ are scales (characteristic values) of the horizontal, vertical, and Coriolis accelerations, respectively. According to (2.2), the dimensionless values (denoted with over-line above) have the order of unity. Thus, for non-extreme climate processes with scales (2.1), the following dimensionless parameters are small

$$\varepsilon \equiv \left(\frac{A_{\mathrm{hor}}}{g}, \frac{A_{\mathrm{ver}}}{g}, \frac{A_{\mathrm{cor}}}{g}\right) \leq 10^{-2}. \qquad (2.4)$$

Thus, the inertial forces are negligibly small compared to gravity.

It makes sense to consider the asymptotics of the equations of hydro- and thermodynamics of the atmosphere $(1.1) - (1.5)$, when

$$\varepsilon \to 0, \qquad (2.5)$$

for decreasing values of $V_{\mathrm{hor}}, V_{\mathrm{ver}}$ and increasing values of $L_{\mathrm{hor}}, L_{\mathrm{ver}}, \tau$. Here, the viscous forces are small and not considered.

## 2.2 Quasi-hydrostatic approximation and evaluation of vertical velocity

For the scales (2.1) and small value of $\epsilon$, from the momentum equations (1.2) – (1.4) we get [1]

$$\frac{1}{\rho g}\frac{\partial p}{\partial x} = O(\varepsilon), \quad \frac{1}{\rho g}\frac{\partial p}{\partial y} = O(\varepsilon), \quad \frac{1}{\rho g}\frac{\partial p}{\partial z} = -1 + O(\varepsilon). \qquad (2.6)$$

As a result, the pressure of air at a certain height is determined by the total mass above in the vertical column:

$$\frac{\partial p}{\partial z} = -\rho g(1 + O(\varepsilon)), \quad p(t,x,y,z) = gM(1 + O(\varepsilon)),$$
$$M(t,x,y,z) = \int\limits_z^\infty \rho(t,x,y,z')\,\mathrm{d}z' \approx \int\limits_z^H \rho(t,x,y,z')\,\mathrm{d}z'. \qquad (2.7)$$

---

[1]Here and below $O(\varphi)$ means, that

$$\lim_{\varphi \to 0}\frac{O(\varphi)}{\varphi} = K, \quad 0 < K < \infty$$





Here $H$ is the height, over which the total mass, $M$, air pressure, $p$, and air density, $\rho$ are negligibly small compared to those on the surface of Earth ($z = 0$). In particular, one can take $H \approx 40$ km, then

$$p(t, x, y, H) \approx 0,003\, p(t, x, y, 0),$$
$$\rho(t, x, y, H) \approx 0,004\, \rho(t, x, y, 0). \tag{2.8}$$

The distribution of vertical velocity is evaluated using the continuity equation (1.1) and thermodynamic energy equation (1.7):

$$\frac{\partial v_z}{\partial z} = -\left(\frac{\partial v_x}{\partial x} + \frac{\partial v_y}{\partial y}\right) - \frac{1}{\rho}\frac{d\rho}{dt} =$$
$$= -\left(\frac{\partial v_x}{\partial x} + \frac{\partial v_y}{\partial y}\right) - \frac{1}{\gamma p}\frac{dp}{dt} + \frac{\gamma - 1}{\gamma}\frac{Q^*}{p}. \tag{2.9}$$

Taking into account the quasi-hydrostatic equation for pressure (2.7), we obtain

$$\frac{dp}{dt} = g\left(\int\limits_z^H \frac{\partial \rho}{\partial t}dz' - \rho v_z\right)(1 + O(\varepsilon)) + v_x\frac{\partial p}{\partial x} + v_y\frac{\partial p}{\partial y}. \tag{2.10}$$

It follows from (1.1) that

$$\int\limits_z^H \frac{\partial \rho}{\partial t}dz' = -\int\limits_z^H \mathrm{div}(\rho\mathbf{v})dz' = \dot{M} + \rho v_z g,$$

$$\dot{M} = -\int\limits_z^H \left(\frac{\partial(\rho v_x)}{\partial x} + \frac{\partial(\rho v_y)}{\partial y}\right)dz. \tag{2.11}$$

Substituting this expression into (2.10), we get

$$\frac{dp}{dt} = g\dot{M} + v_x\frac{\partial p}{\partial x} + v_y\frac{\partial p}{\partial y} + g(\rho v_z + \dot{M})O(\varepsilon). \tag{2.12}$$

As a result, substituting (2.12) into (2.9), we obtain the equation for vertical velocity:

$$\frac{\partial v_z}{\partial z} = -\left(\frac{\partial v_x}{\partial x} + \frac{\partial v_y}{\partial y}\right) - \frac{g\dot{M}}{\gamma p} - \frac{v_x}{\gamma p}\frac{\partial p}{\partial x} - \frac{v_y}{\gamma p}\frac{\partial p}{\partial y}$$

$$+ \frac{\gamma - 1}{\gamma}\frac{Q^*}{p} + g\frac{\dot{M} + \rho v_z}{\gamma p}O(\varepsilon). \tag{2.13}$$

The first two terms on the right side of the equation above are estimated as follows:

$$\frac{\partial v_x}{\partial x}, \frac{\partial v_y}{\partial y} = O\left(\frac{V_{\mathrm{hor}}}{L_{\mathrm{hor}}}\right). \tag{2.14}$$

Using the mean-value theorem, we estimate the third term on the right side of (2.13):

$$\frac{g\dot{M}}{\gamma p} = \frac{\dot{M}}{\gamma M} = \frac{-\int_z^H (\partial(\rho v_x)/\partial x + \partial(\rho v_y)/\partial y)dz'}{\gamma \int_z^H \rho(t, x, y, z')dz'} \sim$$
$$\sim \frac{\hat{\rho} V_{\mathrm{hor}}(H - z)/L_{\mathrm{hor}}}{\gamma \tilde{\rho}(H - z)} = O\left(\frac{V_{\mathrm{hor}}}{L_{\mathrm{hor}}}\right), \tag{2.15}$$





where $\hat{\rho}(z)$ and $\tilde{\rho}(z)$ are the mean densities of the considered integrals over vertical coordinate from $z$ to $H$ in the denominator and numerator, respectively. Here we have $\tilde{\rho}(z)/\hat{\rho}(z) = O(1)$.

To estimate the fourth and fifth terms on the right side of (2.13), the horizontal gradients of pressure is estimated from (2.6) and (2.3)

$$\frac{\partial p}{\partial x}, \frac{\partial p}{\partial y} = \rho O\left(\frac{V_{\text{hor}}^2}{L_{\text{hor}}}\right). \tag{2.16}$$

Then, taking into account (2.15), we get

$$\frac{v_x(\partial p/\partial x) + v_y(\partial p/\partial y)}{g\dot{M}} =$$
$$= \frac{\{v_x(\partial p/\partial x) + v_y(\partial p/\partial y)\}/\gamma p}{g\dot{M}/\gamma p} \sim$$
$$\sim \frac{V_{\text{hor}}\rho}{\gamma p} O\left(\frac{V_{\text{hor}}^2}{L_{\text{hor}}} + \frac{V_{\text{hor}}^2}{L_{\text{cor}}}\right) / O\left(\frac{V_{\text{hor}}}{L_{\text{hor}}}\right) = \tag{2.17}$$
$$= O\left(\frac{V_{\text{hor}}^2}{C^2} + \frac{L_{\text{hor}}V_{\text{hor}}^2}{L_{\text{cor}}C^2}\right) = \left(1 + \frac{L_{\text{hor}}}{L_{\text{cor}}}\right)O\left(\text{Ma}^2\right),$$

where $\text{Ma} \equiv V_{\text{hor}}/C$ is the Mach number of the horizontal motion and $C = (\gamma p/\rho)^{1/2} = 300-350$ m/s is the sound speed. The magnitude of $\text{Ma}^2$ is small under the scales of (2.1). Furthermore, we show that the asymptotics $\text{Ma}^2 \to 0$ as $\epsilon \to 0$ corresponds to the following relation:

$$\text{Ma}^2 = \frac{V_{\text{hor}}^2}{C^2} = \frac{L_{\text{hor}}g}{C^2}\frac{V_{\text{hor}}^2}{gL_{\text{hor}}} = \frac{L_{\text{hor}}g}{C^2}\varepsilon. \tag{2.18}$$

Here we consider that the magnitude of $L_{\text{hor}}$ does not exceed the magnitude of $C^2/g \sim 10^4$ m by orders.

Thus, according to (2.17) and (2.18), the terms of horizontal advection of pressure gradients are negligibly small as $\epsilon \to 0$.

As a result, the equation (2.12) and the differential equation following from it (2.13) take the following form:

$$\frac{\mathrm{d}p}{\mathrm{d}t} = g\dot{M} + g\left(\dot{M} + \rho v_z\right)O(\varepsilon), \tag{2.19}$$

$$\frac{\partial v_z}{\partial z} = -\left(\frac{\partial v_x}{\partial x} + \frac{\partial v_y}{\partial y}\right) - \frac{\dot{M}}{\gamma M} + \frac{\gamma-1}{\gamma}\frac{Q^*}{p} + \frac{\dot{M} + \rho v_z}{\gamma M}O(\varepsilon). \tag{2.20}$$

### 2.3 Asymptotic system of equations in case of small inertial foeces

Given the fields of $\rho$, $v_x$ and $v_y$, the fields of $M$ and $\dot{M}$ can be calculated using (2.7) and (2.11):

$$M(z - \mathrm{d}z) = M(z) + \rho(z)\mathrm{d}z,$$
$$\dot{M}(z - \mathrm{d}z) = \dot{M}(z) - \left(\frac{\partial(\rho v_x)}{\partial x} + \frac{\partial(\rho v_y)}{\partial y}\right)\mathrm{d}z. \tag{2.21}$$





The equation for the vertical velocity $v_z$ in the form

$$\frac{\partial v_z}{\partial z} = -\left(\frac{\partial v_x}{\partial x} + \frac{\partial v_y}{\partial y}\right) - \frac{g\dot{M}}{\gamma p} + \frac{\gamma-1}{\gamma}\frac{Q^*}{p} -$$
$$- \frac{v_x}{\gamma p}\frac{\partial p}{\partial x} - \frac{v_y}{\gamma p}\frac{\partial p}{\partial y} \qquad (2.22)$$

was presented in the book (Lorenz and Lorenz, 1967), but in recent years, it has hardly been mentioned. A similar but different equation for the adiabatic regime ($Q^* = 0$) was also obtained in (Eliassen, 1949). However, these works do not show that the

last two terms of (2.22) are in the order of $\epsilon$, i.e., the inertial forces are negligibly small in comparison with gravity force.

In total, we have the following system of hydro- and thermodynamic equations for the inviscid air in the field of gravity with the quasi-hydrostatic approximation along the vertical coordinate:

$$\frac{\partial \rho}{\partial t} = -v_x\frac{\partial \rho}{\partial x} - v_y\frac{\partial \rho}{\partial y} - v_z\frac{\partial \rho}{\partial z} + \rho\left(\frac{1}{\gamma}\frac{\dot{M}}{M} - \frac{\gamma-1}{\gamma}\frac{Q^*}{gM}\right),$$

$$\rho\frac{\partial v_x}{\partial t} = -\rho v_x\frac{\partial v_x}{\partial x} - \rho v_y\frac{\partial v_x}{\partial y} - \rho v_z\frac{\partial v_x}{\partial z} - g\frac{\partial M}{\partial x} + \rho f v_y,$$

$$\rho\frac{\partial v_y}{\partial t} = -\rho v_x\frac{\partial v_y}{\partial x} - \rho v_y\frac{\partial v_y}{\partial y} - \rho v_z\frac{\partial v_y}{\partial z} - g\frac{\partial M}{\partial y} - \rho f v_x,$$

$$\frac{\partial v_z}{\partial z} = -\left(\frac{\partial v_x}{\partial x} + \frac{\partial v_y}{\partial y}\right) - \left(\frac{1}{\gamma}\frac{\dot{M}}{M} - \frac{\gamma-1}{\gamma}\frac{Q^*}{gM}\right),$$

$$\frac{\partial \dot{M}}{\partial z} = \frac{\partial(\rho v_x)}{\partial x} + \frac{\partial(\rho v_y)}{\partial y},$$

$$\frac{\partial M}{\partial z} = -\rho. \qquad (2.23)$$

Here $f = 2\Omega\sin\theta$ is Coriolis parameter ($\theta$ is the latitude). In this system, the temperature can be calculated from the equation

of state of air:

$$T = \frac{gM}{\rho R}, \quad (p = gM). \qquad (2.24)$$

If the density, $\rho(x,y,z,t)$, and surface pressure, $p(x,y,0,t)$, are known, the pressure, $p(x,y,z,t)$, can be calculated by integration of the quasi-hydrostatic equation (2.7). Then temperature $T(x,y,z,t)$ can be obtained via the state equation (2.24). The vertical velocity, $v_z$, and the pressure, $p$, are inertialess as they are determined by the distributions of density $\rho$, horizontal

velocities, $v_x$, $v_y$, at the same moment of time. Thus, for non-extreme meteorological processes in the atmosphere with scales (2.1), the inertia effects only through the horizontal velocities.

Until now, we have proven the following theorem.

**Theorem 1.** *The equation for vertical velocity* (2.20) *and the entire system of equations* (2.23) *are asymptotically exact for the flow of inviscid air described by Eqs.* (1.1) − (1.5) *if parameter $\epsilon$ (see* (2.4)*), which determines the ratio of the characteristic*

*values of the inertial forces (horizontal, vertical and Coriolis) to the gravity force, approaches to zero (see* (2.5)*).*





## 3 System of equations for perturbations

### 3.1 Dimensionless form of vertical quasi-hydrostatic equations

In addition to (2.2), we introduce the following dimensionless variables:

$$\overline{\rho} = \frac{\rho}{\rho_0}, \quad \overline{M} = \frac{M}{\rho_0 L_{\mathrm{ver}}}, \quad \overline{\dot{M}} = \frac{\dot{M}\tau}{\rho_0 L_{\mathrm{ver}}}, \tag{3.1}$$

where $\rho_0$ is the characteristic density at the surface of the Earth ($z = 0$).

The system of equations (2.23) is non-linear or quasilinear due to the convection terms in continuity equation and horizontal momentum equations. With the use of (3.1) the system (2.23) can be represented in the dimensionless form

$$\mathbf{B}_t \frac{\partial \overline{\mathbf{U}}}{\partial \overline{t}} + \mathbf{B}_x \frac{\partial \overline{\mathbf{U}}}{\partial \overline{x}} + \mathbf{B}_y \frac{\partial \overline{\mathbf{U}}}{\partial \overline{y}} + \mathbf{B}_z \frac{\partial \overline{\mathbf{U}}}{\partial \overline{z}} + \mathbf{B} = 0,$$

$$\overline{\mathbf{U}} = \left( \overline{\rho}, \overline{v}_x, v_y, \overline{v}_z, \overline{\dot{M}}, \overline{M} \right)^{\mathrm{T}} \tag{3.2}$$

where $\overline{\mathbf{U}}$ is a column matrix of dependent variables (superscript T denotes transposed matrix), $\mathbf{B}_t, \mathbf{B}_x, \mathbf{B}_y, \mathbf{B}_z, \mathbf{B}$ are matrices

of coefficients defined by dependent variables in $\overline{\mathbf{U}}$ (see Appendix A).

The derivative $\partial^2 T / \partial z^2$, as a component of heat source $Q^*$ through eddy heat transfer, is not considered as a part of the differential operator (3.2).

In the dimensionless system of equations the following dimensionless parameters are defined:

$$\overline{Q} = \frac{\gamma - 1}{\gamma} \frac{\tau Q^*}{\rho_0 L_{\mathrm{ver}} g}, \quad \overline{g} = \frac{L_{\mathrm{ver}} g}{L_{\mathrm{hor}^2}} \quad \overline{f} = f\tau, \tag{3.3}$$

they determine the external forcings, i.e., heat flux, gravity, and Coriolis force, respectively. If we use the following values for scales:

$$L_{\mathrm{hor}} \sim 10^4\,\mathrm{m}, \quad L_{\mathrm{ver}} \sim 10^3\,\mathrm{m}, \quad \tau \sim 10^3\mathrm{s}, \quad \Delta T \sim 10\mathrm{K}, \tag{3.4}$$

here $\Delta T$ is the characteristic value of temperature change in period $\tau$, then accounting that $\tau Q^* = \rho c_p \Delta T$, we have the following estimations for the dimensionless scales (3.3)

$$\overline{Q} \sim \frac{\gamma - 1}{\gamma} \frac{\rho c_p \Delta T}{p_0} \sim \frac{R \Delta T}{RT} \sim 10^{-1}, \quad \overline{g} \sim 10^2, \quad \overline{f} \sim 10^{-1}. \tag{3.5}$$

### 3.2 Stability of small shortwave perturbation to the original solution

We consider some solution to the system (3.2):

$$\overline{\mathbf{U}} = \overline{\mathbf{U}}(\overline{t}, \overline{x}, \overline{y}, \overline{z}), \tag{3.6}$$

which is named after the *original solution*. And exists another solution $\overline{\mathbf{U}}^{(\mathrm{d})}$ to (3.2) with a small perturbation $\overline{\mathbf{U}}'$ to $\overline{\mathbf{U}}$:

$$\overline{\mathbf{U}}^{(\mathrm{d})} = \overline{\mathbf{U}} + \overline{\mathbf{U}}', \quad \left( |\overline{\mathbf{U}}| \sim 1, |\overline{\mathbf{U}}'| \sim \delta \ll 1 \right) \tag{3.7}$$





Substituting (3.7) into the system (3.2) and subtracting equations (3.2) for the original solution $\overline{\mathbf{U}}$ and neglecting the nonlinear terms of the second and third-order of $\delta$, we obtain a system of quasi-linear differential equations for the perturbation $\overline{\mathbf{U}}'$:

$$
\mathbf{B}_t \frac{\partial \overline{\mathbf{U}}'}{\partial \overline{t}} + \mathbf{B}_x \frac{\partial \overline{\mathbf{U}}'}{\partial \overline{x}} + \mathbf{B}_y \frac{\partial \overline{\mathbf{U}}'}{\partial \overline{y}} + \mathbf{B}_z \frac{\partial \overline{\mathbf{U}}'}{\partial \overline{z}} + \mathbf{B}' \overline{\mathbf{U}}' = \mathbf{F}',
$$
$$
\overline{\mathbf{U}}' = \left( \overline{\rho}', \quad \overline{v}'_x, \quad \overline{v}'_y, \quad \overline{v}'_z, \quad \dot{\overline{M}}', \quad \overline{M}' \right)^{\mathrm{T}}
$$
(3.8)

where $\mathbf{B}'$ as $\mathbf{B}_t, \mathbf{B}_x, \mathbf{B}_y, \mathbf{B}_z$ is a matrix, which consists of parameters depending on the original solution (unperturbed) to the system (3.2), and the column matrix $\mathbf{F}'$ is determined by the perturbation of the heat flux $\overline{Q}'$ (see Appendix B).

We consider small harmonic perturbations as the solution to the system (3.8), which can be represented in the following form with the help of imaginary unit $i = \sqrt{-1}$ and complex variables:

$$
t = 0: \quad \overline{\mathbf{U}}' = \mathbf{A}_* \exp \left( i \overline{k}_x \overline{x} + \overline{k}_y \overline{y} + \overline{k}_z \overline{z} \right) \right),
$$
$$
\mathbf{A}_* = \left( A_*^{(\rho)}, A_*^{(vx)}, A_*^{(vy)}, A_*^{(vz)}, A_*^{(\dot{M})}, A_*^{(M)} \right)^{\mathrm{T}}
$$
(3.9)

where $A_*^{(j)} = A^{(j)} + i A_{**}^{(j)}$ define the complex amplitudes of perturbations of corresponding values ($\left| A^{(j)} \right| \sim \delta \ll 1, j = \overline{\rho}, \overline{v}_x,$ $\overline{v}_y, \overline{v}_z, \dot{\overline{M}}, \overline{M}$).

We assume that the wavelengths of shortwave perturbations are much shorter than the characteristic length scale, and the heat flux perturbation is zero ($\overline{Q}' = 0$):

$$
\overline{k}_x = k_x L_{\mathrm{hor}}, \overline{k}_y = k_y L_{\mathrm{hor}}, \overline{k}_z = k_z L_{\mathrm{hor}} \gg 1, \overline{Q}' = 0,
$$
$$
l'_x = \frac{2\pi}{k_x} \ll L_{\mathrm{hor}}, \quad l'_y = \frac{2\pi}{k_y} \ll L_{\mathrm{hor}}, \quad l'_z = \frac{2\pi}{k_z} \ll L_{\mathrm{ver}}.
$$
(3.10)

For such perturbations, we have $\mathbf{F}' = 0$. The coefficients $\mathbf{B}_t, \mathbf{B}_x, \mathbf{B}_y, \mathbf{B}_z, \mathbf{B}'$ with parameters of the original solution $\overline{\mathbf{U}}$ vary within distances of $L_{\mathrm{hor}}$ and $L_{\mathrm{ver}}$, whch are much larger than the wavelengths of perturbations $l'_x, l'_y, l'_z$. Thus, these matrices of coefficients can be regarded as constants. Then system (3.8) becomes a homogeneous system of linear equations with constant coefficients, and among its solutions for $t > 0$, there exist solutions in the form of traveling wave:

$$
\overline{v}'_x = A_*^{(vx)} \boldsymbol{E}_*, \quad \overline{v}'_y = A_*^{(vy)} \boldsymbol{E}_*, \quad \overline{v}'_z = A_*^{(vz)} \boldsymbol{E}_*,
$$
$$
\overline{\rho}' = A_*^{(p)} \boldsymbol{E}_*, \quad \dot{\overline{M}}' = A_*^{(M)} \boldsymbol{E}_*, \quad \overline{M}' = A_*^{(M)} \boldsymbol{E}_*,
$$
(3.11)

where

$$
\boldsymbol{E}_* = \exp \left( i \left( \overline{k}_x \overline{x} + \overline{k}_y \overline{y} + \overline{k}_z \overline{z} - \overline{\omega}_* \overline{t} \right) \right)
$$
$$
= \exp \left( \overline{\omega}_{**} \overline{t} \right) \cdot \exp \left( i \left( \overline{k}_x \overline{x} + \overline{k}_y \overline{y} + \overline{k}_z \overline{z} - \overline{\omega} \overline{t} \right) \right)
$$
$$
\overline{\omega}_* = \overline{\omega} + i \overline{\omega}_{**}, \quad \overline{\omega}_* = \omega_* \cdot \tau.
$$

Substituting (3.11) into the system of equations (3.8), we obtain a homogeneous linear system of algebraic equations:

$$
\mathbf{D} \mathbf{A}_* = 0, \quad \mathbf{D} = i \left( -\overline{\omega}_* \mathbf{B}_t + \overline{k}_x \mathbf{B}_x + \overline{k}_y \mathbf{B}_y + \overline{k}_z \mathbf{B}_z \right) + \mathbf{B}'.
$$
(3.12)




Therefore, for the homogeneous system of algebraic equations $\mathbf{D}\mathbf{A}_* = 0$ having a nonzero solution, the condition $\det \mathbf{D} = 0$ reduces to a cubic characteristic equation with respect to $\overline{\omega}_*$:

$$\overline{\omega}_*^3 + b_*\overline{\omega}_*^2 + c_*\overline{\omega}_* + d = 0, \tag{3.13}$$

where $b_*, c_*, d_*$ are complex parameters (see Appendix C) defined by the original solution to system (3.6).

Obviously, if among the roots of the characteristic equation (3.13) exists such a root $\overline{\omega}_* = \overline{\omega} + i\overline{\omega}_{**}$ that increment $\overline{\omega}_{**} > 0$, the amplitude of perturbation in (3.11) increases with time. This indicates the instability of the investigated solution (3.6). If at the same time

$$\overline{\omega}_{**} \underset{\overline{k} \to \infty}{\longrightarrow} +\infty, \tag{3.14}$$

then such a solution holds *absolute instability*, and the Cauchy problem of the system (3.2) is ill-posed (Godunov et al., 1976).

In contrast, the negative increment $\overline{\omega}_{**} < 0$ ensures stability. In the case of neutral stability of the linear approximation ($\overline{\omega}_{**} = 0$), the conclusion about the instability of solution (3.6) requires the study of the nonlinear behavior of the system (2.23).

## 220  4   Linear instability of original solutions

### 4.1   Instability of resting-state solution

We consider the regime when there is no motion and heat influx in the original solution, and the density is uniform horizontally:

$$\overline{v}_x = \overline{v}_y = \overline{v}_z = 0, \quad \frac{\partial \overline{\rho}}{\partial \overline{x}} = \frac{\partial \overline{\rho}}{\partial \overline{y}} = 0,$$

$$\frac{\partial \overline{v}_x}{\partial \overline{t}} = \frac{\partial \overline{v}_y}{\partial \overline{t}} = 0, \quad \frac{\partial \overline{v}_x}{\partial \overline{x}} = \frac{\partial \overline{v}_x}{\partial \overline{y}} = \frac{\partial \overline{v}_x}{\partial \overline{z}} = 0, \tag{4.1}$$

$$\frac{\partial \overline{v}_y}{\partial \overline{x}} = \frac{\partial \overline{v}_y}{\partial \overline{y}} = \frac{\partial \overline{v}_y}{\partial \overline{y}} = \frac{\partial \overline{v}_z}{\partial \overline{z}} = 0, \quad \dot{\overline{M}} = 0, \quad \overline{Q} = 0.$$

The horizontal $\overline{k}_{\text{hor}}$ and vertical wavenumbers $\overline{k}_{\text{ver}}$ are defined

$$\overline{k}_{\text{hor}} = \sqrt{\overline{k}_x^2 + \overline{k}_y^2}, \quad \overline{k}_{\text{ver}} = \overline{k}_z. \tag{4.2}$$

With resting-state conditions (4.1), the cubic characteristic equation (3.13) is simplified as

$$\overline{\omega}_* \left[ \overline{\omega}_*^2 - \left( \left( \overline{N}^2 - \overline{G}^2 \right) \frac{\overline{k}_{\text{hor}}^2}{\overline{k}_{\text{ver}}^2} + \overline{f}^2 - \frac{\overline{N}^2 \overline{G}^2}{\overline{g}} \frac{k_{\text{hor}}^2}{\overline{k}_{\text{ver}}^3} \cdot i \right) \right] = 0. \tag{4.3}$$

$$\overline{N}^2 = -\frac{\overline{g}}{\overline{\rho}} \frac{\partial \overline{\rho}}{\partial \overline{z}} = \left( \frac{L_{\text{ver}}\tau}{L_{\text{hor}}} \right)^2 N^2 \quad \left( N^2 = -\frac{g}{\rho} \frac{\partial \rho}{\partial z} \right)$$





$$\overline{G}^2 = \frac{\overline{\rho g}}{\gamma \overline{M}} = \left(\frac{L_{\mathrm{ver}}\tau}{L_{\mathrm{hor}}}g\right)^2 \frac{\rho}{\gamma p} \quad \left(G^2 = \left(\frac{g}{C}\right)^2 = \frac{g^2\rho}{\gamma p}\right)$$

Here $\overline{N}$ is the dimensionless value of the Brent-Väisälä frequency $N$, $C$ is sound speed. For scales (3.4) and standard

atmosphere (Atmosphere, 1976) we have the following estimates:

$$N^2 = -\frac{g}{\rho}\frac{\partial \rho}{\partial z} \sim \frac{g}{L_{\mathrm{ver}}} \sim 10^{-2}\mathrm{s}^{-2}, \quad \overline{N}^2 = \left(\frac{L_{\mathrm{ver}}\tau}{L_{\mathrm{hor}}}\right)^2 N^2 \sim 10^2,$$

$$G^2 = \frac{g^2}{C^2} \sim 10^{-3}\mathrm{s}^{-2}, \quad \overline{G}^2 = \left(\frac{L_{\mathrm{ver}}\tau}{L_{\mathrm{hor}}}\right)^2 G^2 \sim 10^1. \tag{4.4}$$

The dependency of $N^2 - G^2$ on height for the standard atmosphere is presented in Figure 1. It is shown that $N^2 - G^2 \sim 10^{-4}\,\mathrm{s}^{-2}$, and at height $z \approx 4$ km, the sign of $N^2 - G^2$ changes.

For the cubic equation (3.13) despite the root $\overline{\omega}_*^{(1)} = 0$, expressions of the other two roots are shown in Appendix D when

$\overline{k}_x, \overline{k}_y, \overline{k}_z \gg 1$. Among these roots, there always exists one with a positive increment ($\overline{\omega}_{**} > 0$), which indicates the instability of the resting-state solution.

The small Coriolis force

$$\overline{f} \ll \frac{\overline{k}_{\mathrm{hor}}}{\overline{k}_{\mathrm{ver}}}\sqrt{\left|\overline{N}^2 - \overline{G}^2\right|}, \tag{4.5}$$

can take place at near-equatorial latitudes ($\theta \ll 1$) and a height not close to 4 km. In such situation for $\overline{k}_{\mathrm{hor}} \to \infty$,

$$\left(\frac{\left|\overline{N}^2\overline{G}^2\right|}{\overline{g}}\frac{\overline{k}_{\mathrm{hor}}^2}{\overline{k}_{\mathrm{ver}}^3}\right) \Bigg/ \left(\left|\overline{N}^2 - \overline{G}^2\right|\frac{\overline{k}_{\mathrm{hor}}^2}{\overline{k}_{\mathrm{ver}}^2}\right) =$$

$$= \frac{\left|\overline{N}^2\overline{G}^2\right|}{\overline{g}\left|\overline{N}^2 - \overline{G}^2\right|}\frac{1}{\overline{k}_{\mathrm{ver}}} \xrightarrow[\overline{k}_{\mathrm{ver}}\to\infty]{} 0. \tag{4.6}$$

From the general formulae of roots (Appendix D), we obtain the expressions of nonzero roots of (3.13) for small Coriolis force:

$$\overline{\omega}_{**}^{(2,3)} \simeq \pm A_1 \frac{\overline{k}_{\mathrm{hor}}}{\overline{k}_{\mathrm{ver}}^2}, \quad \overline{\omega}^{(2,3)} \simeq \mp B_1 \frac{\overline{k}_{\mathrm{hor}}}{\overline{k}_{\mathrm{ver}}}. \tag{4.7}$$

$$\left(A_1 = \frac{\left|\overline{N}^2\overline{G}^2\right|}{2\overline{g}\sqrt{\left|\overline{N}^2 - \overline{G}^2\right|}}, B_1 = \sqrt{\left|\overline{N}^2 - \overline{G}^2\right|}\right)$$

When $\overline{k}_{\mathrm{hor}} \to \infty$ and $\overline{k}_{\mathrm{ver}} \to \infty$, there are various asymptotics, including

$$\overline{\boldsymbol{\omega}}_{**}^{(2,3)} = \pm A_1\kappa_1 \xrightarrow[\kappa_1\to\infty]{} \pm\infty, \quad \overline{\omega}_{**}^{(2,3)} = \pm A_1\kappa_1 \xrightarrow[\kappa_1\to 0]{} 0. \tag{4.8}$$

$$\left(\kappa_1 = \frac{\overline{k}_{\mathrm{hor}}}{\overline{k}_{\mathrm{ver}}^2}\right)$$



If, in contrast to (4.5), the Coriolis force prevails:

$$\overline{f} \gg \frac{\overline{k}_{\text{hor}}}{\overline{k}_{\text{ver}}} \sqrt{\left| \overline{N}^2 - \overline{G}^2 \right|}, \tag{4.9}$$

which can take place at high latitudes or altitudes near 4 km, then according to Appendix D we get

$$\overline{\omega}_{**}^{(2,3)} \simeq \pm A_2 \kappa_2, \quad \overline{\omega}^{(2,3)} \simeq \mp \overline{f}. \tag{4.10}$$

$$\left( A_2 = \frac{\left| \overline{N}^2 \overline{G}^2 \right|}{2 \overline{g} \overline{f}}, \quad \kappa_2 = \frac{\overline{k}_{\text{hor}}^2}{\overline{k}_{\text{ver}}^3}, \right)$$

As in (4.8) for $\overline{k}_{\text{hor}} \to \infty$ and $\overline{k}_{\text{ver}} \to \infty$, exist different asymptotics, including

$$\overline{\omega}_{**}^{(2,3)} = \pm A_2 \kappa_2 \underset{\kappa_2 \to \infty}{\to} \pm \infty, \quad \overline{\omega}_{**}^{(2,3)} = \pm A_2 \kappa_2 \underset{\kappa_2 \to 0}{\to} 0. \tag{4.11}$$

From (4.8) and (4.11) it follows that the resting-state of the system of hydrodynamic equations with the vertical quasi-hydrostatic approximation (2.23) is unstable, in particular, in case of shortwave perturbations with vertical wavelengths $\overline{l}'_{\text{ver}} = 2\pi/\overline{k}_{\text{ver}}$ longer than horizontal ($\overline{l}'_{\text{hor}} = 2\pi/\overline{k}_{\text{hor}}$, $\overline{k}_{\text{ver}}$, $\overline{k}_{\text{hor}} \gg 1$, and $\overline{k}_{\text{ver}} < \overline{k}_{\text{hor}}$), is absolute unstable. For perturbations when the vertical wavelength is many times shorter than the horizontal wavelength ($\overline{l}'_{\text{ver}} \ll \overline{l}'_{\text{hor}}$, $\overline{k}_{\text{ver}} \gg \overline{k}_{\text{hor}} \gg 1$), the resting-state tends to be neutral stable.

The positive values of the increment of the amplitude of perturbation $\overline{\omega}_{**}$ at resting-state for $\overline{k}_z = 15$ and $\overline{k}_z = 150$ with the horizontal wavelengths $\overline{k}_x = \overline{k}_y = \sqrt{2}/2 \overline{k}_{\text{hor}}$ are shown in Figure 2 by the lines with indicator number 0.

The main conclusion is that the shorter the horizontal wavelengths (the larger $\overline{k}_{\text{hor}} \gg 1$), the faster the amplitude of perturbation at resting-state develops. Moreover, the shorter the vertical wavelengths (the larger $\overline{k}_{\text{ver}} \gg 1$), the more slowly the perturbation at resting-state grows.

## 4.2 Instability of vertical quasi-hydrostatic system with different approximations

Shortwave perturbation at resting-state of the atmosphere with a vertical quasi-hydrostatic approximation is also studied by (Arakawa and Konor, 2009). However, in the original system the following equation (thereinafter, the Arakawa inexact approximationt) is used

$$\frac{\partial M}{\partial t} = \dot{M} \quad \left( \dot{M} = -\int\limits_{z}^{+\infty} \left( \frac{\partial \left( \rho v_x \right)}{\partial x} + \frac{\partial \left( \rho v_y \right)}{\partial y} \right) \mathrm{d}z' \right), \tag{4.12}$$

instead of the asymptotically exact equation (2.19), which is equivalent to

$$\frac{\partial M}{\partial t} = \dot{M} + \rho v_z. \tag{4.13}$$

Note that the equation for perturbation at resting-state corresponding to (4.12) has the form

$$\frac{\partial M'}{\partial t} = \dot{M}'. \tag{4.14}$$





If instead of equation (2.19) or (4.13) the Arakawa inexact approximation (4.12) is adopted, then instead of (3.13) the characteristic equation becomes:

$$\overline{\omega}_* \left[ \overline{\omega}_*^2 - \overline{f}^2 - i\overline{g}\frac{\overline{k}_{\text{hor}}^2}{\overline{k}_{\text{ver}}} \right] = 0. \tag{4.15}$$

Nonzero roots of this equation are determined by the following formulae:

$$\overline{\omega}_{**}^{(2,3)} = \pm \frac{\sqrt{2\overline{g}}}{2}\frac{\overline{k}_{\text{hor}}}{\overline{k}_{\text{ver}}^{1/2}}, \quad \overline{\omega}^{(2,3)} = \mp \frac{\sqrt{2\overline{g}}}{2}\frac{\overline{k}_{\text{hor}}}{\overline{k}_{\text{ver}}^{1/2}}$$
$$\text{if} \quad \overline{f}^2 \ll \overline{g}\overline{k}_{\text{hor}}^2/\overline{k}_{\text{ver}} \tag{4.16}$$

$$\overline{\omega}_{**}^{(2,3)} = 0, \quad \overline{\omega}^{(2,3)} = \pm\overline{f}, \quad \text{if} \quad \overline{f}^2 \gg \overline{g}\overline{k}_{\text{hor}}^2/\overline{k}_{\text{ver}} \tag{4.17}$$

It is clear that these expressions corresponding to the usage of the Arakawa inexact approximation, differ from (4.7) and (4.10). In the case of the predominance of Coriolis force, this difference is fundamental, namely, instead of absolute instability in (4.11), neutral stability follows from (4.17). In addition to the inexact equation (4.12), the conclusion about shortwave instability in (Arakawa and Konor, 2009) is made from the values of the real part of the root $\overline{\omega}$, not the imaginary part $\overline{\omega}_{**}$. Such conclusion is inaccurate, as can be seen from (4.16).

In the case of small Coriolis force, the Arakawa inexact approximation gives absolute instability, with following asymptotics

$$\overline{\omega}_{**} \underset{\kappa_3 \to \infty}{\to} A_3\kappa_3 \to \infty. \quad \left( A_3 = \frac{\sqrt{2\overline{g}}}{2}, \kappa_3 = \frac{\overline{k}_{\text{hor}}}{\overline{k}_{\text{ver}}^{1/2}} \right) \tag{4.18}$$

Now we consider other approximations instead of the asymptotically exact equation for the vertical velocity $v_z$ (2.19) or its equivalence (4.13). The simplest approximation for $v_z$ is zero vertical velocity, in particular, for the perturbation of vertical 290 velocity,

$$v_z' = 0. \tag{4.19}$$

The corresponding characteristic equation at resting-state (4.1) has the form

$$\overline{\omega}_* \left[ \overline{\omega}_*^2 - \overline{f}^2 + \overline{G}^2 \frac{\overline{k}_{\text{hor}}^2}{\overline{k}_{\text{ver}}^2} \right] = 0, \tag{4.20}$$

from which it follows that

$$\overline{\omega}_*^{(1)} = 0, \quad \overline{\omega}_*^{(2,3)} = \pm\sqrt{\overline{f}^2 - \overline{G}^2\frac{\overline{\boldsymbol{k}}_{\text{hor}}^2}{\overline{\boldsymbol{k}}_{\text{ver}}^2}}. \tag{4.21}$$





The root with positive increment $\overline{\omega}_{**}$ when $\overline{f} < \overline{Gk}_{\mathrm{hor}}/\overline{k}_{\mathrm{ver}}$ can lead to absolute instability (3.14). In particular, or small Coriolis force,

$$\overline{\omega}_{**} \underset{\kappa_4 \to \infty}{\to} \overline{G}\kappa_4 \to \infty \left( \kappa_4 = \frac{\overline{k}_{\mathrm{hor}}}{\overline{k}_{\mathrm{ver}}} \right). \tag{4.22}$$

The Holton inexact vertical quasi-hydrostatic approximation (Holton and Hakim, 2012) with equation of the constant local
pressure (thereinafter, the Holton inexact approximation) can also be used:

$$\frac{\mathrm{d}p}{\mathrm{d}t} = -\rho v_z g \quad \left( \dot{M} = -\overline{\rho v}_z, \quad \frac{\partial p}{\partial t} = 0 \right). \tag{4.23}$$

Using this approximation instead of equation (2.19) or its equivalence (4.13), the characteristic equation becomes

$$\overline{\omega}_* \left[ \overline{\omega}_*^2 - \overline{f}^2 - \overline{N}^2 \kappa_4^2 \right] = 0, \tag{4.24}$$

and its roots take the following form:

$\overline{\omega}_*^{(1)} = 0, \quad \overline{\omega}_*^{(2,3)} = \pm\sqrt{\overline{f}^2 + \overline{N}^2 \kappa_4^2}. \tag{4.25}$

It follows that the solution of the resting-state to the system of equations with the Holton inexact vertical quasi-hydrostatic approximation is neutrally stable ($\overline{\omega}_{**}^{(2,3)} = 0$), if

$$\frac{\partial \rho}{\partial z} < 0 \quad \left( \overline{N}^2 > 0 \right). \tag{4.26}$$

Otherwise, when

$\frac{\partial \rho}{\partial z} > 0 \quad \left( \overline{N}^2 < 0 \right), \tag{4.27}$

i.e., the density increases vertically, such original solution of resting-state is unstable for small Coriolis force $\overline{f} < \left| \overline{N}\kappa_4 \right|$, and analogously to (4.22), absolutely unstable.

One can also adopt the vertical quasi-hydrostatic approximation with the quasi-incompressibility

$$\frac{\mathrm{d}\rho}{\mathrm{d}t} = 0, \tag{4.28}$$

from which together with the continuity equation, instead of (2.19) the vertical velocity $v_z$ is obtained by the following equation

$$\frac{\partial v_z}{\partial z} = -\frac{\partial v_x}{\partial x} - \frac{\partial v_y}{\partial y}. \tag{4.29}$$

For such a system , the characteristic equation at resting-state (4.1) has the same form (4.24), as for the Holton inexact approximation (4.23).

The vertical quasi-hydrostatic approximation with constant local density is proposed in (Marchuk, 1974) (thereinafter, the Marchuk inexact approximation):

$$\frac{\partial \rho}{\partial t} = 0. \tag{4.30}$$





Together with the continuity equation, the equation for the vertical velocity $v_z$ is obtained as following

$$\frac{\partial \rho v_z}{\partial z} = -\frac{\partial \rho v_x}{\partial x} - \frac{\partial \rho v_y}{\partial y}.$$ (4.31)

The characteristic equation for this equation at resting-state (4.1) has the form

$$\overline{\omega}_* \left( \overline{\omega}_*^2 - \overline{f}^2 \right) = 0,$$ (4.32)

which corresponds to the neutral stability.

     The Marchuk inexact approximation (4.30) and (4.31), the Arakawa approximation (4.12), the zero velocity approximation (4.19) and the Holton inexact approximation of constant local pressure (4.23) are valid only when the vertical velocity of

air is comparatively small. Only the Marchuk inexact approximation leads to a neutral stable solution at resting-state for all wavenumbers of perturbations.

     The shortwave instability at resting-state is also studied by (Moore, 1985) for a two-dimensional system ($v_y = v_y' = 0$) with approximation (4.29), and taking into account of the perturbation of the heat flux $\overline{Q}' \neq 0$, which is assumed to be proportional to the perturbation of vertical velocity $\overline{v}_z'$. The characteristic equation is close to a specific occasion ($v_y = v_y' = 0, \overline{k}_y = 0$) of

equation (4.24).

     From (4.8), (4.11), (4.18) and (4.22), it can be seen that the asymptotics for the increment of perturbations for various approximations is determined by the coefficients of the relations between $\overline{k}_{\mathrm{hor}}$ and $\overline{k}_{\mathrm{ver}}$, namely

$$\kappa_1 = \frac{\overline{k}_{\mathrm{hor}}}{\overline{k}_{\mathrm{ver}}^2}, \quad \kappa_2 = \frac{\overline{k}_{\mathrm{hor}}^2}{\overline{k}_{\mathrm{ver}}^3}, \quad \kappa_3 = \frac{\overline{k}_{\mathrm{hor}}}{\overline{k}_{\mathrm{ver}}^{1/2}}, \quad \kappa_4 = \frac{\overline{k}_{\mathrm{hor}}}{\overline{k}_{\mathrm{ver}}}.$$

### 4.3   Instability of one-dimensional vertical motion

A one-dimensional vertical atmospheric model is used to analyze physical processes in the formation of climate. In such a case, the distribution of vertical velocity is defined by the heat flux $\overline{Q}'$ and horizontal inflow $\dot{\overline{M}}$, which should be set parametrically, and the perturbation spreads only in the vertical direction:

$$\overline{v}_x = \overline{v}_y = \overline{v}_x' = \overline{v}_y' = 0, \quad \frac{\partial \overline{v}_x}{\partial \overline{t}} = \frac{\partial \overline{v}_y}{\partial \overline{t}} = 0,$$


$$\frac{\partial \overline{v}_x}{\partial \overline{x}} = \frac{\partial \overline{v}_x}{\partial \overline{y}} = \frac{\partial \overline{v}_x}{\partial \overline{x}} = \frac{\partial \overline{v}_y}{\partial \overline{x}} = \frac{\partial \overline{v}_y}{\partial \overline{y}} = \frac{\partial \overline{v}_y}{\partial \overline{z}} = 0.$$ (4.33)

     Thus, the system of equations (2.23) is simplified as

$$\frac{\partial \overline{\rho}}{\partial \overline{t}} = -\overline{v}_z \frac{\partial \overline{\rho}}{\partial \overline{z}} + \overline{\rho} \left( \frac{1}{\gamma} \frac{\dot{\overline{M}}}{\overline{M}} - \frac{\overline{Q}}{\overline{M}} \right),$$

$$\frac{\partial \overline{v}_z}{\partial \overline{z}} = \frac{\overline{Q}}{\overline{M}} - \frac{1}{\gamma} \frac{\dot{\overline{M}}}{\overline{M}},$$

$$\frac{\partial \overline{M}}{\partial \overline{z}} = -\overline{\rho}.$$ (4.34)





Here the value of $\dot{\overline{M}}(t,z)$ should be set manually. In this case, the cubic equation (3.13) degenerates into a linear one, and

its root is

$$
\overline{\omega}_{**} = \left( \frac{\partial \overline{\rho}/\partial \overline{z}}{\overline{M} \overline{k}_z^2} - 1 \right) \left( \frac{\overline{Q}}{\overline{M}} - \frac{\dot{\overline{M}}}{\gamma \overline{M}} \right) = \left( \frac{\partial \overline{\rho}/\partial \overline{z}}{\overline{k}_z^2 \overline{M}} - 1 \right) \frac{\partial \overline{v}_z}{\partial \overline{z}},
$$

$$
\overline{\omega} = \overline{k}_z \overline{v}_z + \frac{\overline{\rho}}{\overline{M}} \left( \frac{\dot{\overline{M}}}{\gamma \overline{M}} - \frac{\overline{Q}}{\overline{M}} \right) \frac{1}{\overline{k}_z} = \overline{k}_z \overline{v}_z - \frac{\overline{\rho}}{\overline{M}} \frac{\partial \overline{v}_z}{\partial \overline{z}} \frac{1}{\overline{k}_z}.
$$
(4.35)

For a standard atmosphere, $\partial \overline{\rho}/\partial \overline{z} < 0$, one always has

$$
\frac{\partial \overline{\rho}/\partial \overline{z}}{\overline{M}} \sim -10^{-2},
$$
(4.36)

from which it follows that

$$
\overline{\omega}_{**} \approx -\frac{\partial \overline{v}_z}{\partial \overline{z}}.
$$
(4.37)

     Thus the shortwave stability for a one-dimensional atmosphere ($\overline{\omega}_{**} < 0$) takes place when the vertical velocity increases

with height ($\partial \overline{v}_z/\partial \overline{z} > 0$), this happens with the presence of heating ($\overline{Q} > 0$) and horizontal outflow above the given height

($\dot{\overline{M}} < 0$). Otherwise, when cooling occurs ($\overline{Q} < 0$) and there is inflow above ($\dot{\overline{M}} > 0$), then the vertical velocity decreases with

height ($\partial \overline{v}_z/\partial \overline{z} < 0$), and such state is unstable ($\overline{\omega}_{**} > 0$).

### 4.4   Instability of solution with motion

Using the roots of the cubic equation (3.13), we study the stability of three-dimensional perturbation to a solution with motion

with equal horizontal wavenumbers. Then according to (4.2):

$$
\overline{k}_x = \overline{k}_y = \frac{\sqrt{2}}{2} \overline{k}_{\text{hor}}
$$
(4.38)

To calculate the roots of (3.13), we set the parameters of the original solution with motion as

$$
\begin{aligned}
\overline{v}_x = \overline{v}_y = \overline{v}_z = 1, &\quad \overline{Q} = 0, \\
\frac{\partial \overline{v}_y}{\partial \overline{x}} = \frac{\partial \overline{v}_x}{\partial \overline{y}} = 0.5, &\quad \frac{\partial \overline{v}_x}{\partial \overline{x}} = \frac{\partial \overline{v}_y}{\partial \overline{y}} = -0.5, \\
\frac{\partial \overline{v}_x}{\partial \overline{z}} = \frac{\partial \overline{v}_y}{\partial \overline{z}} = \frac{\partial \overline{v}_z}{\partial \overline{z}} = 0, &\quad \frac{\partial \overline{\rho}}{\partial \overline{x}} = \frac{\partial \overline{\rho}}{\partial \overline{y}} = 0,
\end{aligned}
$$
(4.39)

the values of $\overline{\rho}, \overline{M}, \partial \overline{\rho}/\partial \overline{z}$ are chosen according to the standard atmosphere (Atmosphere, 1976) at height $z = 10$ km. Values

of these parameters $\partial \overline{\rho}/\partial \overline{t}, \partial \overline{v}_x/\partial \overline{t}, \partial \overline{v}_y/\partial \overline{t}$ and $\dot{\overline{M}}$ are calculated from (2.23).

Numerical values of increments $\overline{\omega}_*^{(j)} (j = 1, 2, 3)$ depending on $\overline{k}_{\text{hor}}$ are shown in Figure 3, they are calculated from the

cubic equation (3.13) for fixed vertical wavelengths $\overline{k}_{\text{ver}} = 15, 50, 150$, and parameters given by the original solution (4.39).

Numerical indicators correspond to the number of roots $j = 1, 2, 3$. It is clear that the positive increment $\overline{\omega}_{**}$ grows with the

increase of $\overline{k}_{\text{hor}}$. It follows from Appendix C that the values $\overline{k}_{\text{hor}}/\overline{k}_{\text{ver}}$ strongly change the coefficients of the cubic equation

(3.13). For an increase of $\overline{k}_{\text{ver}}$, the growth of positive increment $\overline{\omega}_{**}$ greatly weakens.





In Figure 2, in addition to the resting-state (numerical indicator 0), the positive values of increment $\overline{\omega}_{**}$ depending on $\overline{k}_{\text{horr}}, \overline{k}_{\text{ver}}$ for two sets of horizontal velocity are also shown. The lines with indicators 1 and 2 refer to the state with horizontal velocities given in (4.39) and the state with horizontal velocities twice as larger, respectively. The dimensionless horizontal velocity in the order of unity affects the increment for wavenumbers $\overline{k}_{\text{hor}} \sim 10^1$, yet the horizontal velocity does not strongly affect the increment $\overline{\omega}_{**}$ for shortwave perturbations ($\overline{k}_{\text{hor}} > 10^2$).

## 5    Use of pseudo-viscosities to eliminate linear instability

### 5.1    Shear and bulk pseudo-viscosities

This analysis does not consider the influence of boundary conditions. The boundaries can prohibit the energy influx from outside, which contributes to the kinetic energy of perturbations and the growth of perturbation amplitude. The boundary conditions in the framework of the boundary value problem can significantly limit the growth of the perturbation. Nevertheless, 380   the shortwave instability, of course, is one of the reasons of the possible instability of finite-difference calculations.

    While modelling (3.2) by the finite-difference method, small shortwave perturbations can be generated, the minimum wavelength of which $l_i^{\min}$ is defined by the grid size $\Delta_i$:

$$\overline{l}_i^{\min} = 4\overline{\Delta}_i, \quad \overline{k}_i^{\max} \overline{l}_i^{\min} = 2\pi, \quad \overline{k}_i^{\max} = k_i^{\max} L_i = {}^1/{}_2 \pi N_i,$$
$$\left(\overline{\Delta}_i = \Delta\overline{x}, \Delta\overline{y}, \Delta\overline{z}, \quad N_i = L_i/\Delta_i, \quad i = x, y, z\right) \tag{5.1}$$

where $N_x, N_y, N_z$ are the numbers of nodes at characteristic length $L_x, L_y, L_z$, in directions $x, y, z$, respectively.

The terms with turbulent viscosity in the horizontal momentum equations are relatively small in comparison with other terms. However, the pseudo-viscosities, which are much greater than real turbulent viscosity of air, can be added to eliminate the instability due to numerical shortwave perturbations. They influence the horizontal momentum equations of the perturbations, and then change the form of algebraic equations (3.12) through

$$\mu\left(\frac{\partial^2 v_i'}{\partial x^2} + \frac{\partial^2 v_i'}{\partial y^2} + \frac{\partial^2 v_i'}{\partial z^2}\right), \lambda\frac{\partial}{\partial x_i}\left(\frac{\partial v_x'}{\partial x} + \frac{\partial v_y'}{\partial y} + \frac{\partial v_z'}{\partial z}\right) \rightarrow$$
$$\mu\left\{A^{(vi)}\overline{k}_x^2 + A^{(vi)}\overline{k}_y^2 + A^{(vi)}\overline{k}_z^2\right\},$$
$$\lambda\overline{k}_i\left\{A^{(vx)}\overline{k}_x + A^{(vy)}\overline{k}_y + A^{(vz)}\overline{k}_z\right\}, \tag{5.2}$$
$$(i = x, y, z)$$

where $\mu$ and $\lambda$ are coefficients of shear and bulk pseudo- viscosities, respectively. We set the dimensionless values of these coefficients growing with increasing wavenumbers:

$$\overline{\mu} = c_\mu \overline{\mu}_a \left(\overline{k}_x \overline{k}_y \overline{k}_z\right)^n, \quad \overline{\lambda} = c_\lambda \overline{\mu}_a \left(\overline{k}_x \overline{k}_y \overline{k}_z\right)^m$$
$$\overline{\mu}_a = \frac{\mu_a}{\rho V_{\text{ver}} L_{\text{ver}}} \quad \left(\mu_a = 1.8 \times 10^{-5} \text{kg/(m·s)}\right) \tag{5.3}$$

Here $\mu_a$ is the viscosity of the air.





## 5.2 Influence of pseudo-viscosities for resting-state solution

Similar as (3.13) the characteristic equation for a resting-state solution, taking into account the shear pseudo-viscosity in accordance with (5.2), turns out to be

$$\overline{\omega}_*^3 + 2i\overline{\mu}\overline{k}^2\overline{\omega}_*^2 - \left[\overline{\mu}^2\overline{k}^4 + \alpha + i\cdot\beta\right]\overline{\omega}_*$$
$$+\overline{\mu}\left[\beta + i\left(\overline{f}^2 - \alpha\right)\right]\overline{k}^2 = 0, \tag{5.4}$$

$$\alpha = \left(\overline{N}^2 - \overline{G}^2\right)\frac{\overline{k}_{\text{hor}}^2}{\overline{k}_{\text{ver}}^2} + \overline{f}^2, \quad \beta = -\frac{\overline{N}^2\overline{G}^2}{\overline{g}}\frac{\overline{k}_{\text{hor}}^2}{\overline{k}_{\text{ver}}^3},$$

$$\overline{k}^2 = \overline{k}_x^2 + \overline{k}_y^2 + \overline{k}_z^2 = \overline{k}_{\text{hor}}^2 + \overline{k}_{\text{ver}}^2$$

A similar cubic equation for a restiing-state solution with bulk pseudo-viscosity $\lambda$ has the following form

$$\overline{\omega}_*\left[\overline{\omega}_*^2 - \frac{\overline{\lambda G}^2\overline{k}_{\text{hor}}^2}{\overline{g}\overline{k}_{\text{ver}}}\overline{\omega}_* - (\alpha + i\cdot\beta)\right] = 0. \tag{5.5}$$

It is clear that for $m > 0$ (see (5.3)) and $\overline{k}_i \gg 1$ $(i = x, y, z)$, we have

$$c_\lambda\overline{\mu}_a\left(\overline{k}_x\overline{k}_y\overline{k}_z\right)^m\frac{\overline{G}^2}{\overline{g}}\frac{\overline{k}_{\text{hor}}^2}{\overline{k}_{\text{ver}}} \gg |\beta| = \left|\frac{\partial\overline{\rho}}{\partial\overline{z}}\right|\frac{\overline{G}^2}{\overline{\rho}}\frac{\overline{k}_{\text{hor}}^2}{\overline{k}_{\text{ver}}^3}, \tag{5.6}$$

then from the characteristic equation (5.5), we obtain

$$\overline{\omega}_{**}^{(1)} = 0, \quad \left|\overline{\omega}_{**}^{(2,3)}\right| \xrightarrow[\overline{k}_{\text{hor}}\to\infty]{} 0 \tag{5.7}$$

In Figure 4, the values of increment $\overline{\omega}_{**}^{(j)}(j = 1, 2, 3)$ with the existence of shear pseudo-viscosity at the resting solution (4.1) are shown. The values $\overline{\omega}_{**}$ in dependency on $\overline{k}_{\text{hor}}$ are calculated for $\overline{k}_{\text{ver}} = 15$ and different coefficients of shear pseudo-viscosity $n = 1.5, \quad c_\mu = 0, 1, 10$. The effect of bulk pseudo-viscosity $\overline{\lambda}$ on stability at resting regime does not qualitatively differ from the effect of shear pseudo-viscosity $\overline{\mu}$.

However, an increase in pseudo-viscosity can not achieve the point that all three roots $(\overline{\omega}_*^{(j)}(j = 1, 2, 3))$ of the characteristic

equation have negative increments $\overline{\omega}_{**}^{(j)} < 0$ $(j = 1, 2, 3)$ so that the shortwave perturbation damps and disappears eventually. One of the roots always has a positive increment $\overline{\omega}_{**}^{(3)} > 0$.

## 5.3 Influence of pseudo-viscosities for the solution with motion

For moving air, the pseudo-viscosities can not suppress the global shortwave perturbations of solutions to the asymptotically exact system of equations (2.23) with vertical quasi-hydrostatic approximation. This can be explained by the fact that the

perturbations occupy the entire space, i.e., the perturbations are global and have infinite energy. Real perturbations from the numerical calculations on the difference grids are local and have finite energy. Such shortwave perturbations should be suppressed by pseudo-viscosities (see section 5.4).





Now we consider the influence of pseudo-viscosities (here we consider shear pseudo-viscosity) on the stability for an original solution with motion using the Marchuk inexact approximation. Similarly to (3.13), we obtain the following characteristic
equation:

$$\overline{\omega}_* \left( \overline{\omega}_*^2 + b_1(\overline{\mu})\overline{\omega}_* + c_1(\overline{\mu}) \right) = 0, \tag{5.8}$$

where $b_1(\overline{\mu})$ and $c_1(\overline{\mu})$ are the complex parameters determined by the original solution and shear pseudo-viscosity $\overline{\mu}$. The first root corresponds to neutral linear stability ($\overline{\omega}_{**}^{(1)}$), and the rest two roots correspond to linear stability, i.e., for $\overline{k}_i \to +\infty (i = x, y, z)$, it follows $\overline{\omega}_{**}^{(2,3)} \to -\infty$ (see Figure 5). That is to say, the solution to the system with the Marchuk inexact
approximation for moving air can be unstable, but for shortwave perturbations with the existence of pseudo-viscosity, it turns to be stable. It means that the shear pseudo-viscosity $\overline{\mu}$ (by analogy, as well as $\overline{\lambda}$) can suppress the global shortwave perturbations, and this is the advantage of the Marchuk inexact approximation (4.30) and (4.31) in comparison with the exact approximation.

### 5.4 Influence of local perturbations on instability

Instead of perturbations (3.11), whose amplitude is identical in space, we consider such perturbations that damp in space for
$x, y, z > 0$ with complex wavenumbers $\overline{k}_{j*} = \overline{k}_j + i\overline{k}_{j**}$ $(j = x, y, z)$:

$$\begin{aligned}
\boldsymbol{E}_* &= \exp(i(\overline{\boldsymbol{k}}_{xx}\overline{\boldsymbol{x}} + \overline{\boldsymbol{k}}_{y*}\overline{\boldsymbol{y}} + \overline{\boldsymbol{k}}_{z*}\overline{z} - \overline{\omega}_*\overline{\boldsymbol{t}})) = \\
&= \exp\left( \overline{\omega}_{**}\overline{t} - \overline{k}_{x**}\overline{x} - \overline{k}_{y**}\overline{y} - \overline{k}_{z**}\overline{z} \right) \cdot \\
&\quad \cdot \exp\left( i\left( \overline{k}_x\overline{x} + \overline{k}_y\overline{y} + \overline{k}_z\overline{z} - \overline{\omega}\overline{t} \right) \right)
\end{aligned} \tag{5.9}$$

In Figure 6 the values of increment $\overline{\omega}_{**}^j$ $(j = 1, 2, 3)$ for resting-state solution (4.1) are shown for $\overline{k}_{x**} = \overline{k}_{y**} = \overline{k}_{z**} = 0$ and 20, with $\overline{k}_{\mathrm{ver}} = 15, n = 1.5, c_\mu = 1$. The thin lines correspond to $\overline{k}_{j**} = 0$, and thick lines $\overline{k}_{j**} = 20$. Numerical pointors indicate the root number $j = 1, 2, 3$. It is shown that the shear pseudo-viscosity can stabilize the local shortwave perturbations
$(\overline{k}_x, \overline{k}_y, \overline{k}_z \gg 30)$ for $x, y, z > 0$, since all roots have negative increments $\overline{\omega}_{**}^j < 0$ $(j = 1, 2, 3)$.

It should be noted that for large values $\overline{k}_{j**} \gg 1$, there are large gradients of perturbation parameters (like impulse), and this leads an absolute instability (when $\overline{k}_{j**} \to \infty$) in response to relatively longer wavelength range ($\overline{k}_{\mathrm{hor}} < 30$, for fixed $\overline{k}_{\mathrm{ver}} = 15$). In addition, the amplitude of initial perturbation (5.9) increases for $x, y, z < 0$.

### 6 Conclusions

In this paper, an equation for vertical velocity refines and simplifies the equation obtained by Lorenz and Lorenz (1967). Different from the approximations used in other works such as (Holton and Hakim, 2012; Marchuk, 1974), this equation is asymptotically exact for the system of hydrodynamic equations with small inertia forces compared to the gravity force. The advantage of this system is the absence of sound waves. Indeed, when the density field, $\rho(x, y, z, t)$, is given, the pressure field, $p$, can be calculated without time stepping via the quasi-hydrostatic equation. Then having $\rho$ and $p$, the temperature field, $T$,





is calculated through the equation of state. The absence of sound waves allows for much larger time steps in modelling due to the absence of the Courant-number restriction in the vertical dimension.

     In our analysis the influence of boundary conditions on the instability problem is excluded. Though the boundary conditions in the framework of the boundary value problem can significantly limit the growth of the perturbation, the shortwave instability is one of the reasons of the possible instability of finite-difference calculations.

As a result, the solution to the system of atmospheric dynamics equations with vertical quasi-hydrostatical approximation and without viscosity is known to be unstable for shortwave perturbations occupying the entire space. Even the resting-state solution owns the shortwave instability. The increment number of the perturbation growth relates to the ratio of horizontal to vertical wavenumber of the shortwave perturbations. Such shortwave instability eventually results in the instability introduced by numerical truncation and finite-difference approximation of the derivatives error. It shows that not only usual solutions presenting three-dimensional flow patterns but even the resting-state solution is unstable under shortwave perturbations. More-

over, the larger the ratio of horizontal wavenumber $\overline{k}_{\mathrm{hor}}$ to vertical wavenumber $\overline{k}_{\mathrm{ver}}$ (or the larger the ratio of vertical grid size $\overline{\Delta}_{\mathrm{ver}}$ to horizontal grid size $\overline{\Delta}_{\mathrm{hor}}$), the larger the increment growth of the amplitude of perturbations. Particularly, an infinitely large ratio of vertical grid size to horizontal grid size leads to the absolute instability. Thus, when decreasing the horizontal grid size to achieve better accuracy, one should also decrease the vertical grid size to keep small the ratio $\kappa_2 = \overline{k}_{\mathrm{hor}}^2 / \overline{k}_{\mathrm{ver}}^3 \ll 1$,

e.g., $\overline{\Delta}_{\mathrm{ver}}^3 / \overline{\Delta}_{\mathrm{hor}}^2 \ll 1$.

     For the one-dimensional vertical motion (one-column model) in the standard atmosphere where $\partial\rho/\partial z < 0$, the shortwave stability depends on the vertical velocity profile, $v_z(z)$ or, more specifically, on the sign of $\partial v_z/\partial z$, which relates to the heat input, $Q$, and horizontal mass inflow, $\dot{M}$. Then, due to the heating and horizontal outflow above the given position the condition of the shortwave stability is $\partial v_z/\partial z > 0$.

The pseudo-viscosities taken proportional to the wavenumber of perturbations reduce the increment in the amplitudes of perturbation, so that numerical solutions to the asymptotically exact quasi-hydrostatic system become more stable. In this context, *global perturbations* have non-negative increment causing numerical instability. However, in the case of only *local perturbations*, implementation of pseudo-viscosities allows for making all increments negative, thus yielding practically sound stable solutions.

At the large ratio of the vertical to horizontal grid size, the solution with motion is also unstable for other (inexact) vertical quasi-hydrostatic approximations, namely, those with constant local density (Marchuk, $\partial\rho/\partial t = 0$), constant local pressure (Holton, $\partial p/\partial t = 0$), or quasi-incompressibility ($d\rho/dt = 0$). In contrast to other known differential operators, the inexact vertical quasi-hydrostatic system with constant local density (Marchuk, $\partial\rho/\partial t = 0$) assures the neutral stability for the resting-state solution. And introducing pseudo-viscosities can suppress global perturbations for the solution with motion.

The quasi-hydrostatic equations of the present paper will be beneficial, in particular, for Earth-system modelling with regards to chemical or biogeochemical chains causing the formation of anthropogenic or organic aerosols, condensation nuclei, clouds and precipitation, and electrical charging around water drops. The result about shortwave instability can guide the choice of vertical and horizontal grid sizes for modelling, and proper usage of pseudo-viscosity can reduce the predictability problem caused by shortwave perturbations during calculation.





*Code availability.* The code of this paper can be downloaded by link https://doi.org/10.5281/zenodo.3831455.

*Data availability.* There is no data relevant to the paper.

*Author contributions.* Robert Nigmatulin led the research and provided the main ideas of this paper. Xiulin Xu conducted mathematical deriviations and numerical calculations. Both authors contributed in writing the manuscript.

*Competing interests.* The authors declare that they have no conflict of interest.

*Acknowledgements.* The authors acknowledge the support from Grant of the Ministry of Education and Science of Russian Federation grant No.14.W03.31.0006 through Robert Nigmatulin, and the Chinese Scholarship Council No.201306840046 through Xiulin Xu; and the collaboration with Institute of Atmospheric and Earth-System Research (University of Helsinki) and Finnish Meteorological Institute in frames of the Academy of Finland grant 314 798/799 through Sergej Zilitinkevich.

## Appendix A: Matrices of Coefficients of the Original System of Equations in Dimensionless Form (3.2)

$$
\overline{\mathbf{U}} = \begin{bmatrix} \overline{p} \\ \overline{v}_x \\ \overline{v}_y \\ \overline{v}_z \\ \overline{M} \\ \overline{M} \end{bmatrix} \qquad \mathbf{B} = \begin{bmatrix} \overline{\rho}(\overline{Q}\gamma - \dot{\overline{M}})/(\gamma\overline{M}) \\ -\overline{v}_y\overline{f} \\ \overline{v}_x\overline{f} \\ -(\overline{Q}\gamma - \overline{M})/(\gamma\overline{M}) \\ 0 \\ \overline{\rho} \end{bmatrix}
$$

$$
\mathbf{B}_t = \begin{bmatrix} 1 & 0 & 0 & 0 & 0 & 0 \\ 0 & 1 & 0 & 0 & 0 & 0 \\ 0 & 0 & 1 & 0 & 0 & 0 \\ 0 & 0 & 0 & 0 & 0 & 0 \\ 0 & 0 & 0 & 0 & 0 & 0 \\ 0 & 0 & 0 & 0 & 0 & 0 \end{bmatrix}
$$





$$\mathbf{B}_x = \begin{bmatrix} \overline{v}_x & 0 & 0 & 0 & 0 & 0 \\ 0 & \overline{v}_x & 0 & 0 & 0 & \overline{g}/\overline{\rho} \\ 0 & 0 & \overline{v}_x & 0 & 0 & 0 \\ 0 & 1 & 0 & 0 & 0 & 0 \\ -\overline{v}_x & -\overline{\rho} & 0 & 0 & 0 & 0 \\ 0 & 0 & 0 & 0 & 0 & 0 \end{bmatrix}$$

$$\mathbf{B}_y = \begin{bmatrix} \overline{v}_y & 0 & 0 & 0 & 0 & 0 \\ 0 & \overline{v}_y & 0 & 0 & 0 & 0 \\ 0 & 0 & \overline{v}_y & 0 & 0 & \overline{g}/\overline{\rho} \\ 0 & 0 & 1 & 0 & 0 & 0 \\ -\overline{v}_y & 0 & -\overline{\rho} & 0 & 0 & 0 \\ 0 & 0 & 0 & 0 & 0 & 0 \end{bmatrix}$$

$$\mathbf{B}_z = \begin{bmatrix} \overline{v} & 0 & 0 & 0 & 0 & 0 \\ 0 & \overline{v}_z & 0 & 0 & 0 & 0 \\ 0 & 0 & \overline{v}_z & 0 & 0 & 0 \\ 0 & 0 & 0 & 1 & 0 & 0 \\ 0 & 0 & -\overline{\rho} & 0 & 0 & 0 \\ 0 & 0 & 0 & 0 & 0 & 1 \end{bmatrix}$$

**Appendix B: Matrices of Coefficients of System of Equations for Perturbation in Dimensionless Form** (3.8)

$$\overline{\mathbf{U}}' = \begin{bmatrix} \overline{\rho}' \\ \overline{v}'_x \\ \overline{v}'_y \\ \overline{v}' \\ \frac{M}{M}' \\ \overline{M}' \end{bmatrix} \quad \mathbf{F}' = \begin{bmatrix} -\overline{\rho}\,\overline{Q}'/\overline{M} \\ 0 \\ 0 \\ \overline{Q}'/\overline{M} \\ 0 \\ 0 \end{bmatrix}$$

$$\mathbf{B}' = \begin{bmatrix} B_{11} & \frac{\partial \overline{\rho}}{\partial \overline{x}} & \frac{\partial \overline{\rho}}{\partial \overline{y}} & \frac{\partial \overline{\rho}}{\partial \overline{z}} & \frac{\overline{\rho}}{\gamma \overline{M}} & -\frac{\overline{\rho} B_{11}}{\overline{M}} \\ B_{21} & \frac{\partial \overline{v}_n}{\partial \overline{x}} & \frac{\partial \overline{v}_x}{\partial \overline{y}} - \overline{f} & \frac{\partial \overline{v}_x}{\partial \overline{z}} & 0 & 0 \\ B_{31} & \frac{\partial \overline{v}_0}{\partial \overline{x}} + \overline{f} & \frac{\partial \overline{v}_y}{\partial \overline{y}} & \frac{\partial \overline{v}_y}{\partial \overline{z}} & 0 & 0 \\ 0 & 0 & 0 & 0 & \frac{1}{\gamma \overline{M}} & \frac{B_{11}}{\overline{M}} \\ B_{51} & -\frac{\partial \overline{\rho}}{\partial \overline{x}} & -\frac{\partial \overline{\rho}}{\partial \overline{y}} & 0 & 0 & 0 \\ 1 & 0 & 0 & 0 & 0 & 0 \end{bmatrix}$$





$$B_{11} = \frac{\overline{Q}}{\overline{M}} - \frac{\dot{\overline{M}}}{\gamma \overline{M}}, \quad B_{21} = \frac{1}{\overline{\rho}} \left( \frac{\mathrm{d}\overline{v}_x}{\mathrm{d}\overline{t}} - \overline{v}_y \overline{f} \right),$$

$$B_{31} = \frac{1}{\overline{\rho}} \left( \frac{\mathrm{d}\overline{v}_y}{\mathrm{d}\overline{t}} + \overline{v}_x \overline{f} \right), \quad \frac{\mathrm{d}}{\mathrm{d}\overline{t}} = \frac{\partial}{\partial \overline{t}} + \overline{v}_x \frac{\partial}{\partial \overline{x}} + \overline{v}_y \frac{\partial}{\partial \overline{y}} + \overline{v}_z \frac{\partial}{\partial \overline{z}}$$

$$B_{51} = -\frac{\partial \overline{v}_x}{\partial \overline{x}} - \frac{\partial \overline{v}_y}{\partial \overline{y}}$$

**Appendix C: Complex Coefficients of the Cubic Equation** (3.13)**:**

The following coefficients $b_*$, $c_*$ and $d_*$ are expressed for $\overline{k}_x, \overline{k}_y, \overline{k}_z \gg 1$.

$\quad \mathrm{Re}\, b_* = -3 \left( \overline{k}_x \overline{v}_x + \overline{k}_y \overline{v}_y + \overline{k}_z \overline{v}_z \right) + \mathcal{O}\left( 1/\overline{k}_z \right)$

$$\mathrm{Im}\, b_* = -\frac{\overline{k}_x}{\overline{k}_z} \left( \frac{\partial \overline{v}_x}{\partial \overline{z}} + \frac{\overline{\rho v}_x}{\gamma \overline{M}} \right) - \frac{\overline{k}_y}{\overline{k}_z} \left( \frac{\partial \overline{v}_y}{\partial \overline{z}} + \frac{\overline{\rho v}_y}{\gamma \overline{M}} \right) - B_{51}$$

$$- B_{11} + \mathcal{O}\left( \frac{1}{\overline{k}_z} \right)$$

$$\mathrm{Re}\, c_* = 3 \left( \overline{k}_x \overline{v}_x + \overline{k}_y \overline{v}_y + \overline{k}_z \overline{v}_z \right)^2 + \mathcal{O}(1)$$

$\quad \mathrm{Im}\, c_* = 2 \left( \overline{k}_x \overline{v}_x + \overline{k}_y \overline{v}_y + \overline{k}_z \overline{v}_z \right) \left( B_{11} + \frac{\overline{\rho}}{\gamma \overline{M}} \cdot \frac{\overline{k}_x \overline{v}_x + \overline{k}_y \overline{v}_y}{\overline{k}_z} \right) +$

$$+ 2 \left( \overline{k}_x \overline{v}_x + \overline{k}_y \overline{v}_y \right) \cdot \left( \frac{\overline{k}_x}{\overline{k}_z} \frac{\partial \overline{v}_x}{\partial \overline{z}} + \frac{\overline{k}_y}{\overline{k}_z} \frac{\partial \overline{v}_y}{\partial \overline{z}} \right) +$$

$$+ 2\overline{k}_x \left( \frac{\partial \overline{v}_x}{\partial \overline{z}} \overline{v}_z - \frac{\partial \overline{x}}{\partial \overline{x}} \overline{v}_x - \frac{\partial \overline{v}_y}{\partial \overline{y}} \overline{v}_x \right) +$$

$$+ 2\overline{k}_y \left( \frac{\partial \overline{v}_y}{\partial \overline{z}} \overline{v}_z - \frac{\partial \overline{v}_x}{\partial \overline{x}} \overline{v}_y - \frac{\partial \overline{v}_y}{\partial \overline{y}} \overline{v}_y \right) +$$

$$+ 2\overline{k}_z \overline{v}_z B_{51} + \mathcal{O}\left( \frac{1}{\overline{k}_z} \right)$$


$$\mathrm{Re}\, d_* = -\left( \overline{k}_x \overline{v}_x + \overline{k}_y \overline{v}_y + \overline{k}_z \overline{v}_z \right)^3 + \mathcal{O}\left( \overline{k}_z \right)$$

$$\mathrm{Im}\, d_* = -\left( \overline{k}_x \overline{v}_x + \overline{k}_y \overline{v}_y + \overline{k}_z \overline{v}_z \right)^2 \cdot \mathrm{Z} + \mathcal{O}(1)$$

$$\mathrm{Z} = B_{11} + \frac{\overline{\rho}}{\gamma \overline{M}} \left( \frac{\overline{k}_x}{\overline{k}_z} \overline{v}_x + \frac{\overline{k}_y}{\overline{k}_z} \overline{v}_y \right) + \frac{\partial \overline{v}_x}{\partial \overline{z}} \frac{\overline{k}_x}{\overline{k}_z} + \frac{\partial \overline{v}_y}{\partial \overline{z}} \frac{\overline{k}_y}{\overline{k}_z} + B_{51}$$

**Appendix D: Expressions for the roots of the characteristic equation** (4.3)**:**

$\quad \overline{\omega}_{**}^{(2,3)} = \pm \frac{\sqrt{2}}{2} \left( \sqrt{\alpha^2 + \left( \frac{\overline{N}^2 \overline{G}^2}{\overline{g}} \frac{\overline{k}_{\mathrm{hor}}^2}{\overline{k}_{\mathrm{ver}}^3} \right)^2} - \alpha \right)^{1/2}$



$$\overline{\omega}^{(2,3)} = \mp \frac{\sqrt{2}}{2} \left( \sqrt{\alpha^2 + \left( \frac{\overline{N}^2 \overline{G}^2}{\overline{g}} \frac{\overline{k}^2_{\mathrm{hor}}}{\overline{k}^3_{\mathrm{ver}}} \right)^2} + \alpha \right)^{1/2}$$





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

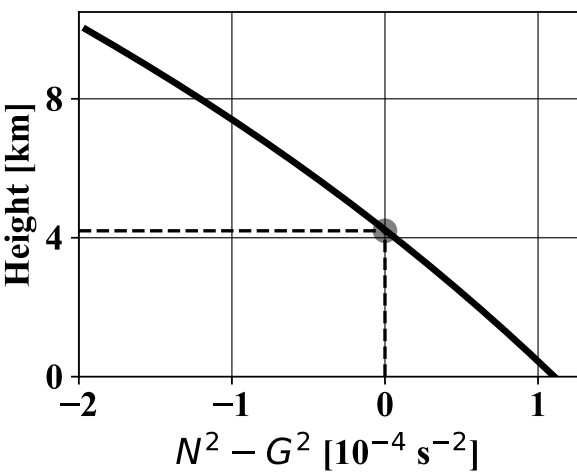

**Figure 1.** value of $N^2 - G^2$ for a standard stable atmosphere

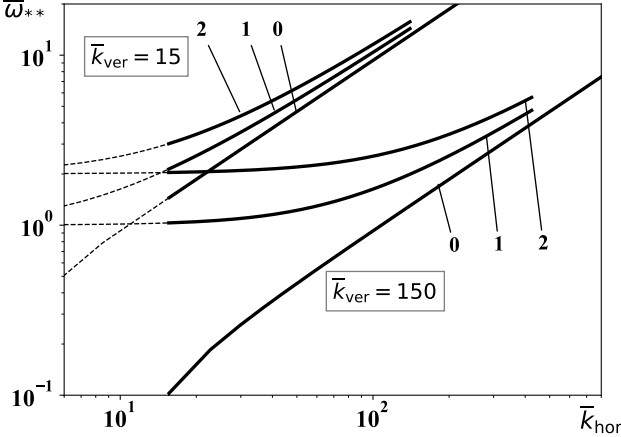

**Figure 2.** dependence of maximum (positive) increment $\overline{\omega}_{**}$ on $\overline{k}_{\mathrm{hor}}$ for fixed vertical wavenumbers $\overline{k}_{\mathrm{ver}} = 15$ and 150, numerical indicator 0 - for resting-state (4.1); indicator 1 - for the original solution (4.39); indicator 2 - for the original solution of which velocity is twice as (4.39).



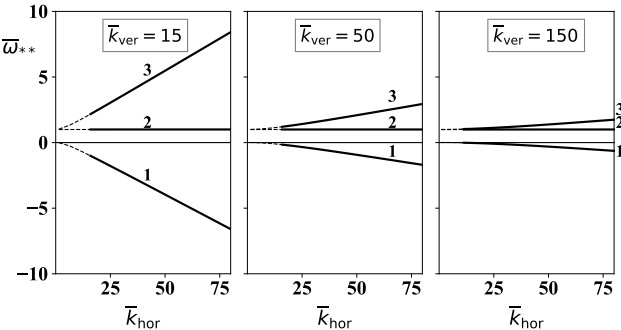

**Figure 3.** dependence of three amplitude increments $\overline{\omega}_{**}^{(j)}$ on $\overline{k}_{\mathrm{hor}}$ for fixed vertical wavenumbers, $\overline{k}_{\mathrm{ver}} = 15$, 50 or 150, for the original solution of (4.39); numerical indicators 1,2,3 correspond to $j = 1, 2, 3$.

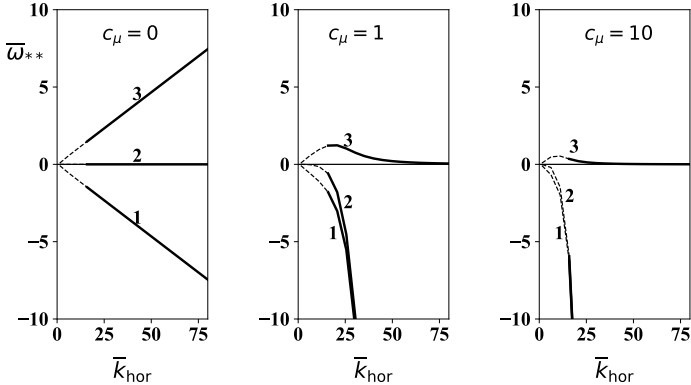

**Figure 4.** dependence of three amplitude increments $\overline{\omega}_{**}^{(j)}$ on $\overline{k}_{\mathrm{hor}}$ for fixed vertical wavenumbers $\overline{k}_{\mathrm{ver}} = 15$; for original solution of resting-state (4.1); different viscosity coefficients $c_\mu = 0$, 1, or 10, and $n = 1.5$ (5.3); numerical indicators 1,2,3 correspond to $j = 1, 2, 3$.

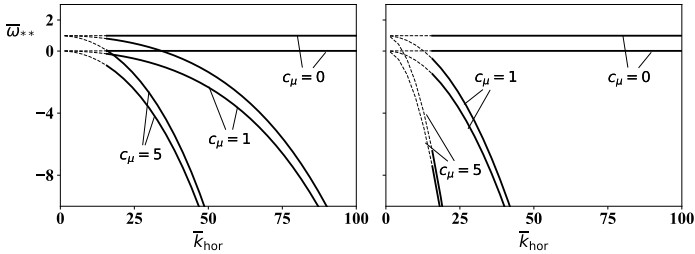

**Figure 5.** dependence of two non-neutral increments $\overline{\omega}_{**}^{(2,3)}$ on $\overline{k}_{\mathrm{hor}}$ for fixed vertical wavenumbers $\overline{k}_{\mathrm{ver}} = 10$ (left) and 20 (right); for the original solution with motion (4.39); different viscosity coefficients $c_\mu = 0$, 1, and 5.

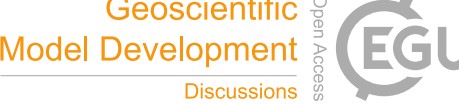

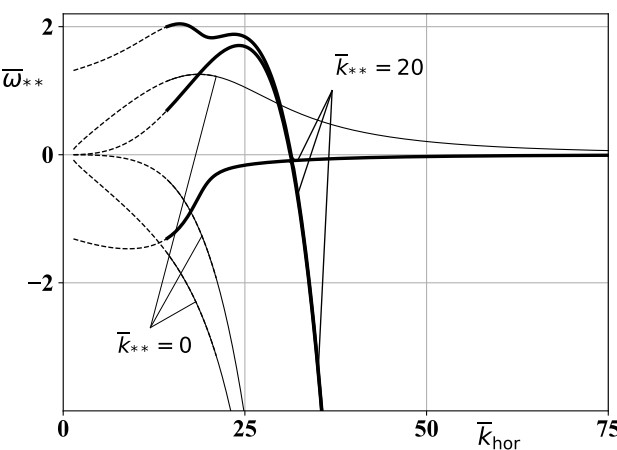

**Figure 6.** dependence of three amplitude increments $\overline{\omega}_{**}$ on $\overline{k}_{\mathrm{hor}}$ for fixed vertical wavenumbers $\overline{k}_{\mathrm{ver}} = 15$; with constant viscosity coefficient $c_\mu = 1$; for the original solution of resting-state (4.1); for different spatial increments $\overline{k}_{**}$, the thin lines correspond to $\overline{k}_{**} = 0$ and the thick lines $\overline{k}_{**} = 20$.