# Peer review of "Quasi-hydrostatic equations for climate models and the study on linear instability"

_Geoscientific Model Development, 2020_

## Short Comment (SC1) · 8 Aug 2020

I lost my original comment here so will keep it short. Try doing some realistic comparisons to data as I did with QBO, see Fig 1 and the following citation

Mathematical Geoenergy, Wiley/AGU (2018) https://agupubs.onlinelibrary.wiley.com/doi/10.1002/9781119434351.ch11

No progress has been made since Holton/Lindzen did the DiffEq formulation years ago, and all you are doing is going deeper and deeper without a simplifying ansatz.

[Figure]

**Fig. 1.**

---

## Author Comment (AC1) · 8 Aug 2020

Dear Paul Pukite,

Thank you very much for your comments.

This paper is devoted to studying the stability property of the system of partial differential equations (with quasi-hydrostatic approximation). We use the linearized equations (not the original equations) to describe the shortwave perturbations. Such perturbations occur in simulations regardless of what numerical scheme is adopted, and grid sizes of meshing determine the (minimum) wavelengths of perturbations. As we do not

develop any numerical methods for solving the original equations, we lack a comparison of the calculation data with real data. But we find out how grid sizes affect the growth rate of small shortwave perturbations in general.

We note that the harmonics in the citation (Mathematical Geoenergy, Wiley/AGU (2018) https://agupubs.onlinelibrary.wiley.com/doi/10.1002/9781119434351.ch11) is different from that of in our paper.

In the citation above, the harmonic external forcing leads to a harmonics-form solution to the original equations. While in our paper, we assume a harmonics-like solution to the system of linearized equations with regards to perturbations.

Best regards,

Xiulin Xu

---

## Short Comment (SC2) · 26 Aug 2020

Dear Dr. Putike,

Actually it is a kind of simplification of GCMs, which were still used in 1980s, with altitude as the vertical coordinate.

But the 'simplification' is made in such a way that additional (uncompensated) vertical velocity appears and conservation of the full energy is violated. Scales (2.1) and the new theorem is another story.

Holton, actually Charney could be the first who expressed the idea, gave a simple

approximate (even in frames of hydrostatic approximation) expression for the vertical velocity, which worked fairy well.

But L.Richardson in 1922 gave an **exact** expression for vertical velocity in frames of hydrostatic approximation. It is just a simple expression of $\frac{\partial v_z}{\partial z}$ from continuity equation, followed by substitution of $\frac{d\rho}{dt}$ by $\frac{dp}{dt}$ with the help of thermodynamic equality.

The 'simplification' made by the First Author of the Paper is the neglect of the horizontal advection of pressure in $\frac{dp}{dt}$ upon scale analyses of **one component** of the horizontal divergence of mass flux. But he did not take into account the very well known caveats of the scale analyses in stratified flows and experience of Margules, Richardson, Charney etc., who discussed specific problems with the calculation of the horizontal divergence in the atmosphere. Sometimes $\frac{dp}{dt}$ may be almost zero, with pressure advection and divergence of mass flux compensating each other. If one drops one them, the balance is violated.

I suppose that if the approach of the Paper would be realized numerically, the simulation would just diverge, so the comparison you've asked would not be even possible.

The Second Author of the paper made a careful linear investigation of the system given to him by the First Author up to the section 3.1. Unfortunately the system in 3.1 is not suitable for simulation of the Earth atmosphere at **any** scale. I would suggest the Second Author to publish a single-author paper on linear analysis as an elaboration of the Arakawa's work, it will be a pity if the careful techniques would be in wane.

My detailed comments on the model by the First Author were submitted to the journal of the original publication and are openly available:

https://pubpeer.com/publications/446D764678B603CC6EF997C8C5EF00#2

https://arxiv.org/pdf/2001.08637.pdf

Best, Ilias Sibgatullin

---

## Short Comment (SC3) · 27 Aug 2020

Dear Ilias Sibgatullin,

Thank you very much for your comments. In the following sentence:

"But the 'simplification' is made in such a way that additional (uncompensated) vertical velocity appears and conservation of the full energy is violated."

Can you explicitly explain how the conservation of full energy is violated in the equations?

To answer

"The 'simplification' made by the First Author of the Paper is the neglect of the horizontal advection of pressure in $\frac{\mathrm{d}p}{\mathrm{d}t}$ upon scale analyses of one component of the horizontal divergence of mass flux.",

I copy the text (slightly different from the current version of manuscript because of written errors) from the paper to show how the horizontal pressure divergence terms are estimated:

To estimate the fourth and fifth terms on the right side of (2.13), the horizontal gradients of pressure is estimated from (2.6) and horizontal momentum equations (1.2), (1.3)

$$\frac{\partial p}{\partial x}, \frac{\partial p}{\partial y} = \rho O\left(\frac{V_{\text{hor}}^2}{L_{\text{hor}}} + \frac{V_{\text{hor}}^2}{L_{\text{cor}}}\right), \quad (L_{\text{cor}} = V_{\text{hor}}\tau_{\text{cor}}). \tag{1}$$

Then, taking into account (2.15), we get

$$\frac{v_x(\partial p/\partial x) + v_y(\partial p/\partial y)}{g\dot{M}} = \frac{\{v_x(\partial p/\partial x) + v_y(\partial p/\partial y)\}/\gamma p}{g\dot{M}/\gamma p} \sim$$

$$\sim \frac{V_{\text{hor}}\rho}{\gamma p} O\left(\frac{V_{\text{hor}}^2}{L_{\text{hor}}} + \frac{V_{\text{hor}}^2}{L_{\text{cor}}}\right)/O\left(\frac{V_{\text{hor}}}{L_{\text{hor}}}\right) = O\left(\frac{V_{\text{hor}}^2}{C^2} + \frac{L_{\text{hor}}V_{\text{hor}}^2}{L_{\text{cor}}C^2}\right) = \left(1 + \frac{L_{\text{hor}}}{L_{\text{cor}}}\right) O\left(\text{Ma}^2\right)$$

Therefore, the horizontal pressure divergence terms were estimated in the frame of scales we are interested in. We can drop these terms in the equation for vertical velocity (2.13) and get the asymptotically exact result for $\epsilon \to 0$ only when we use the scales (2.1).

Thank you again for your interest in our work!

Sincerely,

Xiulin Xu
* * *

---

## Short Comment (SC4) · 27 Aug 2020

Dear Xiulin, you did not show up the scale analyses of the horizontal divergence, I was writing about.

Let's look at it carefully, namely eq. 2.15 (in picture below this text) and put here again the scales of the horizontal divergence, that were used in it:

$$\text{div}_{hor}(\rho\vec{v}) = \frac{\partial \rho v_x}{\partial x} + \frac{\partial \rho v_y}{\partial y} \approx \hat{\rho}\frac{V_{hor}}{L_{hor}}$$

This estimation is *just wrong*, because the components of the divergence compensate each other to large extent.

Want an example?

Take a layer of incompressible fluid and push on its surface with your palm from above. The water will escape from below your palm, and you if you estimate the divergence with only one component it will have a finite value, but of course the full divergence just exactly 0, since the components of the divergence are mutually compensated.

The components of the horizontal divergence in atmosphere also compensate each other to large extent.

I understand that you are in no way responsible for this analyses, since it was present earlier in a paper by the First Author

https://doi.org/10.1134/S0015462818040201

I was pointing at these inconsistencies in this paper already for two years. And now I am amazed to see in again here.

So my comments (attached) were addressed not to you, but to the author of this analysis, Dr. Nigmatulin. The comments await for the answer at the public review portal

https://pubpeer.com/publications/446D764678B603CC6EF997C8C5EF00#2
https://arxiv.org/pdf/2001.08637.pdf

The interest in this analyses is warmed up by the fact that, due to the inconsistencies, the new theorem (new asymptotics) denies the existence of convection and internal waves, which also may have small vertical acceleration. So if such a wrong theory is accepted by a faculty of a university for teaching, it becomes not only wrong but harmful.

$$\frac{g\dot{M}}{\gamma p} = \frac{\dot{M}}{\gamma M} = \frac{-\int_z^H (\partial(\rho v_x)/\partial x + \partial(\rho v_y)/\partial y)\,\mathrm{d}z'}{\gamma \int_z^H \rho(t,x,y,z')\,\mathrm{d}z'} \sim$$

$$\sim \frac{\hat{\rho} V_{\mathrm{hor}}(H-z)/L_{\mathrm{hor}}}{\gamma \tilde{\rho}(H-z)} = O\left(\frac{V_{\mathrm{hor}}}{L_{\mathrm{hor}}}\right),$$

(2.15)

**Fig. 1.**

**On hydrostatic approximation by R.I. Nigmatulin and L.F. Richardson's equation.**

I.N. Sibgatullin[1,*]

[1]*Shirshov Institute of Oceanology of Russian Academy of Sciences*

The theorem given in "Equations of Hydro-and Thermodynamics of the Atmosphere when Inertial Forces Are Small in Comparison with Gravity" (2018) is wrong. The scales given in the paper are not suitable for application of hydrostatic (quasistatic) approximation. The modification of Richardson's equation for vertical velocity by neglecting horizontal advection of pressure results in violation of symmetry of equations and incorrect uncompensated additional vertical velocity.

Keywords: *hydrostatic approximation, quasistatic approximation, synoptic scales, microscale meteorology, mesoscale meteorology, force of inertia*

**I.   INCORRECTNESS OF THE THEOREM FORMULATED IN [1], AND SCALES OF APPLICABILITY OF HYDROSTATIC (QUASISTATIC) APPROXIMATION**

Traditional asymptotic analysis of hydrostatic approximation for different geophysical flows is based on the smallness of the ratio $\varepsilon = H/L$ of the vertical to the horizontal scales of motion, which is often introduced as the *hydrostatic parameter* (see, f.e., R. Zeytounian, [2], eq. 3.9, 19.2). The author of [1] proposed a different approach, which is based only on the smallness of the vertical acceleration normalized by the gravity acceleration, as the parameter of applicability of the hydrostatic approximation, and formulated it as a theorem. Smallness of the amplitude of oscillations, which corresponds to the smallness of the acceleration at a fixed frequency, usually can imply only linearization of the equations. But until now, no one could formulate a theorem on elimination of the short-wave motions and application of the hydrostatic approximation (which is a long-wave approximation according to the traditional asymptotic analysis) based only on the smallness of amplitude or the vertical acceleration.

The transition to the limit from the full Navier−Stokes equations to the equations of hydrostatic approximation when *the vertical acceleration approaches zero does not exist*, since this is directly contradicted by the *finite vertical acceleration of the nontrivial solutions of hydrostatic (quasistatic) approximation equations, in which the equation with the vertical acceleration is replaced by the hydrostatic balance*. In [1] the author also did not give a proof of the existence of such a transition. Therefore, in view of the indicated contradiction, the theorem *formulated in [1] is not true.*

The author applies his own or unusual terminology ("inertialess vertical velocity", "climatic scales"), which is usually not necessary. However, here is the source of confusion and the reason for
* * *
*Electronic address: sibgat@ocean.ru

**Fig. 2.**

---

## Short Comment (SC5) · 27 Aug 2020

It is very nice to see the lively discussion occurring on this manuscript. Can I please remind commentators to constrain their comments to discussion of the material presented.

Manuscripts represent a collective submission by the authors, and author responses should be read as being agreed by the authors unless otherwise stated in the response. Accordingly, please don't attribute perceived issues to individual authors or suggest that individual authors should take particular actions.

Instead the comments should focus on what may or may not be incorrect or otherwise contentious in the science.

Sincerely,

David Ham Executive Editor

---

## Referee Comment (RC1) · Anonymous Referee #1 · 8 Sep 2020

Review of "Quasi-hydrostatic equations for climate models and the study on linear instability" by Robert Nigmatulin and Xiulin Xu

This paper analyzes the sensitivity of the quasi-hydrostatic approximation to the atmospheric fluid equations to differences in horizontal and vertical wavenumber, and to different approximations to the diagnostic equation for vertical motion. In general the math appears to be sound, although as the derivations are not extensive, I have put some faith in the authors that their work is correct. I have instead focused my review primarily on the framing of this derivation and the conclusions drawn from these results. Several comments are as follows:

Line 19: Please contrast the quasi-hydrostatic equations against other systems with filtered sound waves, e.g. anelastic and pseudo-incompressible equations. Also it would be advantageous to give other solutions for avoiding the problem of sound waves, such as implicit-explciit temporal integrators.

Line 21: "Most global climate models are based on a system of dynamic equations in quasi-hydrostatic approximation." I believe this statement to be incorrect. Most global atmospheric models either use the fully non-hydrostatic equations or the hydrostatic equations with shallow atmosphere approximation. To the best of the reviewers knowledge only the UK Met Office model has an option for the quasi-hydrostatic equations. See, for example, Ullrich et al. (2017).

Line 36: "Almost the entire mass of the atmosphere is located in the layer with thickness H of order 10km. So the atmospheric dynamics outside polar zones can be considered in quasi-Cartesian coordiante system..." Why does the relative thinness of the atmosphere affect the use of a Cartesian coordinate system?

Line 36-38: The authors should be more clear that they are using a planar approximation of the equations. Otherwise there isn't an explanation for neglecting the curvature terms in equations (1.2)-(1.4) below.

Line 138: Also see Kasahara and Washington (1966). You may also wish to refer to DeMaria (1995) equation (2.13), which is an example of recent mention of this equation.

Line 140: There has been some work recently showing that these non-hydrostatic terms may be more important in a moist context. See, for example, Gao et al. (2017) or Yang et al. (2017).

Line 145 (equation 2.23): Do the quasi-hydrostatic equations here satisfy any sort of energy principle? If the system exhibits instability for certain ratios of horizontal and vertical grid spacing, then the diagnostic vertical velocity equation must be responsible

for the addition of energy to the system. Presumably one should be able to show which terms are responsible for this violation.

Line 220: It would be advantageous to show how the quasi-hydrostatic equations diverge from the unapproximated equations when it comes to instability. One should be able to show agreement between the quasi-hydrostatic equations and unapproximated equations for a certain regime of k_hor and k_ver

Line 460: If I'm understanding the authors correctly, this instability is present regardless of the values of kappa2. Even for small values of kappa2 the system will eventually go unstable without some external control. So wouldn't a better solution be to use an equation set that actually satisfies a closed energy principle?

Line 475: Can anything be said about the accuracy of these equations, analogous to Davies et al. (2003)?

References:

Davies, T., Staniforth, A., Wood, N. and Thuburn, J., 2003. Validity of anelastic and other equation sets as inferred from normal‐mode analysis. Quarterly Journal of the Royal Meteorological Society: A journal of the atmospheric sciences, applied meteorology and physical oceanography, 129(593), pp.2761-2775.

DeMaria, M., 1995. Evaluation of a hydrostatic, height-coordinate formulation of the primitive equations for atmospheric modeling. Monthly weather review, 123(12), pp.3576-3589.

Gao et al., 2017. Sensitivity of U.S. summer precipitation to model resolution and convective parameterizations across grayÂăzone resolutions. Journal of Geophysical Research: Atmospheres, 122(5), pp.2714-2733.

Kasahara, A. and Washington, W.M., 1966. NCAR global general circulation model of the atmosphere. National Center for Atmospheric Research. http://opensky.ucar.edu/islandora/object/manuscripts%3A859/datastream/PDF/download/citation.pdf

Ullrich, P.A., Jablonowski, C., Kent, J., Lauritzen, P.H., Nair, R., Reed, K.A., Zarzycki, C.M., Hall, D.M., Dazlich, D., Heikes, R., Konor, C. and others, 2017. DCMIP2016: a review of non-hydrostatic dynamical core design and intercomparison of participating models. Geoscientific Model Development, 10, pp.4477-4509.

Yang, Q., Leung, L.R., Lu, J., Lin, Y.L., Hagos, S., Sakaguchi, K. and Gao, Y., 2017. Exploring the effects of a nonhydrostatic dynamical core in high‐resolution aquaplanet simulations. Journal of Geophysical Research: Atmospheres, 122(6), pp.3245-3265.

---

## Author Comment (AC2) · 14 Sep 2020

Dear Sir or Madam:

Thank you very much for reviewing our manuscript. In the following text, we hope to answer the questions raised in this review. Comments of Referee #1 are in italic; our answers are in regular letters.

*Line 19: Please contrast the quasi-hydrostatic equations against other systems with fil- tered sound waves, e.g. anelastic and pseudo-incompressible equations. Also it would be advantageous to give other solutions for avoiding the problem of sound waves, such*

[Figure]

*as implicit-explicit temporal integrators.*

**Answer:** One can use the pseudo-incompressible approximation ($div \overrightarrow{u} = 0$) or the anelastic approximation ($div \rho \overrightarrow{u} = 0$) to filter sound waves by neglecting the time differential term of density in the continuity equation, but the conservation of mass is violated and replaced by the "conservation of volume". In comparison, the quasi-hydrostatic approximation filters the sound wave by neglecting the inertial terms in the vertical momentum equation. The comparison of acoustic-wave-filtered systems is also made in Durran (2008).

The quasi-hydrostatic approximation is valid for long-term and large-scale processes, if we take the vertical length scale to 1 km, and the time scale to 1 hour:

$\frac{\partial v_z}{\partial t} \sim \frac{L_z}{\tau^2} \sim \frac{10^3}{3600^2} \ll g \ (9.8 \frac{m}{s^2})$.

Even if we include the term $\partial v_z / \partial t$ in the vertical momentum equation, the numerical error would be compatible with its real value.

Besides, in Davies (2003) it writes,"Anelastic equation sets are the principal basis of many theoretical and modeling studies of small-scale dynamics, for which they play an analogous role to that of the hydrostatic primitive equations for planetary-scale dynamics."

In fully compressible (unapproximated) equations, one can avoid the problem of sound waves by numerical techniques, like the variational approach Rõõm (1998), implicit or semi-implicit time stepping Tanguay (1990).

*Line 21: "Most global climate models are based on a system of dynamic equations in quasi-hydrostatic approximation." I believe this statement to be incorrect. Most global atmospheric models either use the fully non-hydrostatic equations or the hydrostatic equations with shallow atmosphere approximation. To the best of the reviewers knowledge only the UK Met Office model has an option for the quasi-hydrostatic equations. See, for example, Ullrich et al. (2017).*

**Answer:** We would modify this statement to "The quasi-hydrostatic approximation is adopted in the atmospheric models for studying global long-term non-extreme climatic processes.". The reason why we focus on the hydrostatic equations is that in non-hydrostatic equations, an additional time spacing restriction should be added in the vertical direction:

$\Delta t < K \frac{\Delta z}{V_{ver}}$.

As $\Delta z \ll \Delta x, \Delta y$, much smaller time steps for non-hydrostatic equations are needed.

However, non-hydrostatic effects may be important for processes of smaller scales, for instance, when the cloud microphysics is considered.

*Line 36: "Almost the entire mass of the atmosphere is located in the layer with thickness H of order 10km. So the atmospheric dynamics outside polar zones can be considered in quasi-Cartesian coordinate system..." Why does the relative thinness of the atmosphere affect the use of a Cartesian coordinate system?*

**Answer:** The continuity equation for the one-dimensional spherically symmetric motion (purely radial flow) in the spherical coordinate writes:

$\frac{\partial \rho}{\partial t} + \frac{\partial \rho v}{\partial r} + \frac{2 \rho v}{r} = 0$.

In the case of $\Delta r = H \ (\sim 10 \ km) \ll r (\sim \ 6000 \ km)$, the third term of the continuity equation is negligibly small in comparison to the second term.

*Line 36-38: The authors should be more clear that they are using a planar approximation of the equations. Otherwise there isn't an explanation for neglecting the curvature terms in equations (1.2)-(1.4) below.*

**Answer:** The curvature terms (like the third term $\frac{\rho v}{r}$ in the equation above) can be neglected, since the thickness of the atmosphere is much less than the Earth's radius and then the curvature terms are negligibly small in comparison to the gradient terms.

*Line 138: Also see Kasahara and Washington (1966). You may also wish to refer to*

*DeMaria (1995) equation (2.13), which is an example of recent mention of this equation.*

**Answer:** Different from (2.13) in Kasahara and Washington (1966), we evaluate each term in equation (2.13) of current work and get an asymptotic equation (2.20) for $\varepsilon \to 0$.

In DeMaria (1995), equation (2.13) and thermodynamic equation (2.4) are used as diagnostic equations. Instead of these two equations, in the equation set of the current work, we adopt the equation of vertical velocity (2.20) and the vertical hydrostatic equation (last equation of 2.23), then the temperature is obtained by the state equation (2.24).

*Line 140: There has been some work recently showing that these non-hydrostatic terms may be more important in a moist context. See, for example, Gao et al. (2017) or Yang et al. (2017).*

**Answer:** We totally agree with referee #1 that the non-hydrostatic terms should be taken into account in the models with moist, because the time scale for microphysics of clouds is much smaller than the time scale in the current work, and the vertical velocity of air can influence the diffusion of water vapor. In our model moist is not considered by far.

*Line 145 (equation 2.23): Do the quasi-hydrostatic equations here satisfy any sort of energy principle? If the system exhibits instability for certain ratios of horizontal and vertical grid spacing, then the diagnostic vertical velocity equation must be responsible for the addition of energy to the system. Presumably one should be able to show which terms are responsible for this violation.*

**Answer:** The thermodynamic equation (1.6) in current work is identical with equation (2.4) in DeMaria (1995). It is not included as a diagnostic equation in current work, but the vertical velocity (2.20) is obtained by the thermodynamic equation (1.6) and the continuity equation (1.1). The vertical velocity is not neglected in the horizontal momentum equations. Therefore, the thermodynamic equation (1.6) is valid while using (2.20).

Besides, the energy density takes the form

$e = \frac{v_x^2}{2} + \frac{v_y^2}{2} + \frac{v_z^2}{2} + c_p T,$

If we take the scales: $V_{hor} \sim 10\frac{m}{s}, V_{ver} \sim 1\frac{m}{s}, \ c_p \sim 10^3 \ J/(kg \ K), \ \Delta T \sim 10 \ K$, then the kinetic energy is negligibly small, $e_k \ll e$. Despite this fact, in the current work, the horizontal velocity is obtained by the momentum equations, and the vertical velocity is evaluated by (2.20).

*Line 220: It would be advantageous to show how the quasi-hydrostatic equations diverge from the unapproximated equations when it comes to instability. One should be able to show agreement between the quasi-hydrostatic equations and unapproximated equations for a certain regime of k_hor and k_ver*

**Answer:** As we apply only the quasi-hydrostatic approximation, the unapproximated equations are identical to the Navier-Stocks equations, which are hyperbolic and automatically stable under shortwave perturbations.

*Line 460: If I'm understanding the authors correctly, this instability is present regardless of the values of kappa2. Even for small values of kappa2 the system will eventually go unstable without some external control. So wouldn't a better solution be to use an equation set that actually satisfies a closed energy principle?*

**Answer:** As described in (3.11), the shortwave instability is associated with the positive increment of perturbation amplitude $\overline{\omega}_{**}$, in particular, the higher value of $\overline{\omega}_{**}$ corresponds with stronger instability. As shown in Figure 2, k_ver = 15 corresponds with a higher value of kappa2; thus, in such a case it is more unstable as $\overline{\omega}_{**}$ is larger in comparison with the case k_ver = 150. Also, small positive value of kappa2 leads to small value of $\overline{\omega}_{**}$, it is easy to eliminate such instability using pseudo viscosity introduced in current work or other techniques introduced in Ullrich et al. (2017).

*Line 475: Can anything be said about the accuracy of these equations, analogous to Davies et al. (2003)?*

**Answer:** Equations (2.1) – (2.4) with hydrostatic approximation in Davies et al. (2003) are equivalent to the second, third, sixth and fourth equations of (2.23) in the current work, respectively. As temperature and density are linked by the state equation, in our set of equations, the continuity equation (first equation of (2.23)) is used instead of the thermodynamic equation (2.5) in Davies et al. (2003). Another difference is that we introduce a new variable $\dot{M}$ (analogous to $\frac{D\pi}{Dt}$ in Davies et al. (2003)) for the closure of the equations set.

In terms of accuracy of these equations, we can thus conclude as in Davies et al. (2003), that the hydrostatic equations misrepresent the vertical modes at small horizontal scale. But such a problem does not exist for large horizontal scale.

We will definitely change the manuscript according to your comments and suggestions.

Thank you again for your precious time in reviewing our manuscript!

Sincerely,

Robert Nigmatulin and Xiulin Xu

References

Rõõm, R., 1998: Acoustic Filtering in Nonhydrostatic Pressure Coordinate Dynamics: A Variational Approach. J. Atmos. Sci., **55**, 654–668, https://doi.org/10.1175/1520-0469(1998)055%3c0654:AFINPC%3e2.0.CO;2.

Durran, D. R. (2008). A physically motivated approach for filtering acoustic waves from the equations governing compressible stratified flow. Journal of Fluid Mechanics, 601, 365.

Tanguay, M., Robert, A., & Laprise, R. (1990). A semi-implicit semi-lagrangian fully

compressible regional forecast model. Monthly Weather Review, 118(10), 1970-1980.

---

## Short Comment (SC6) · 15 Sep 2020

Dear David,

I will try to avoid any suggestions. I've put them to soften the remark, and to find some positive moments in the manuscript, which is otherwise a set of mathematical absurdities.

The major incorrectness in the paper is the theorem, which states that as the vertical acceleration approaches zero, the hydrostatic approximation (which admits the finite vertical velocity) is asymptotically exact. A proof of this controversial statement is not

given in the paper. This statement is responsible for the strange scales, for which hydrostatic approximation is applied in the paper.

Now I will point to just another two incorrectness, due to which the system of equations is also incorrect for weather prediction at *any* scales. I would be happy if you join the discussion.

The Authors take the traditional L.F.Richardson's framework (Kasahara 1966) without citation and simplify the expression for the total time derivative of pressure $\frac{dp}{dt}$ by omitting the horizontal pressure advection:

$$p = g \int_z \rho dz \tag{1}$$

$$\frac{dp}{dt} = \frac{\partial p}{\partial t} + \vec{v}\nabla p \tag{2}$$

$$\frac{dp}{dt} = g \int_z \frac{\partial \rho}{\partial t} dz + \vec{v}_{hor}\nabla_{hor}p - v_z\rho g \tag{3}$$

$$\frac{dp}{dt} = g \int_z [-div_{hor}(\rho\vec{v})]dz + v_z\rho g + \vec{v}g \int_z \nabla_{hor}\rho dz - v_z\rho g \tag{4}$$

$$\frac{dp}{dt} = g \int_z [-div_{hor}(\rho\vec{v})]dz + g\vec{v}_{hor} \int_z \nabla_{hor}\rho dz \tag{5}$$

So the Authors neglect the second term in the expression above, based on the very strange scale analysis:
[Figure]

1. They estimate the scale of the divergence of first term by the scale of only one component. But the divergence itself is by order(s) of magnitude less than scale its components.

2. They estimate the $\nabla_{hor}p$ *dynamically* as $U^2/L$. This estimation could be correct if for example we had initially a layer at the state of rest, and then the large horizontal scale perturbation of pressure would produce the waves (propagating with about the speed of sound). But this estimation is absolutely incorrect for the real atmosphere which is set in motion after cyclogenesis with zonal winds and motion of weather fronts. Indeed, the pressure advection can be sometimes very small, as well as the divergence, or sometimes the total derivative of pressure is small. But for a general case the neglect of pressure advection in favor of the divergence of mass flux may result in accumulation of additional vertical velocity.

The pressure advection is very cheap from the computations point of view, if we know the pressure and velocity, but if there was a meaningful reason to get rid of it, the much less ambiguous way would be just to further expand the expressions above:

$$\frac{dp}{dt} = -g \int_z \rho \, div_{hor}(\vec{v}_{hor})dz - g \int_z \vec{v}_{hor} \nabla_{hor}\rho dz + g\vec{v}_{hor} \int_z \nabla_{hor}\rho dz \qquad (6)$$

The 2-nd and 3-d term compensate each other if velocity does not depend on $z$. So this expression

$$\frac{dp}{dt} = -g \int_z \rho \, div_{hor}(\vec{v}_{hor})dz \qquad (7)$$

is exact if $v_{hor}$ does not depend on $z$, for such a case it expresses an obvious fact that

$$\frac{dp}{dt} = -p \, div_{hor}(\vec{v}_{hor}),$$

when the pressure is the height of the air column. And it can be a fair approximation, *in contrast to omitting the pressure advection, which is not actually an approximation but an unbalancing of the expression for the total derivative of pressure.*

I will illustrate it with a wagon of sand (see the attached picture) which corresponds to the horizontal bulk transport of the masses of air. Of course you will never see such a pure motion in the real atmosphere but here I separate it to illustrate the formula.

The pressure corresponds to the height of the sand, and obviously, when the wagon moves as a whole $\frac{dp}{dt} = 0$ for any column of sand. So in the expression of the total derivative $\frac{dp}{dt} = \frac{\partial p}{\partial t} + v\frac{\partial p}{\partial x} = 0$ and $\frac{\partial p}{\partial t} = -v\frac{\partial p}{\partial x}$.

Now if we "neglect" the pressure advection $v\frac{\partial p}{\partial x}$ we will get some finite, and may be very big value of $\frac{dp}{dt}$, instead of ZERO.

Returning to the atmosphere, *this is why the additional vertical velocity may be accumulated*, and *this is why the energy conservation is violated*, since $\frac{dp}{dt}$ is a part of the expression for the vertical velocity and change of the full energy.

I hope that the Authors will answer this detailed claim for the mistakes, or finally retract all these absurdities.
* * *
[Figure]

**Fig. 1.**

---

## Short Comment (SC7) · 17 Sep 2020

Dear Authors,

For your convenience I've regrouped the equations and put the numbers. You may find easier to answer the quations in this message.

The major incorrectness in the paper is the theorem, which states that as the vertical acceleration approaches zero, the hydrostatic approximation (which admits the finite vertical velocity) is asymptotically exact. A proof of this controversial statement is not given in the paper. This statement is responsible for the strange scales, for which

hydrostatic approximation is applied in the paper.

Now I will point to just another two incorrectness, due to which the system of equations is also incorrect for weather prediction at *any* scales.

The Authors take the traditional L.F.Richardson's framework (Kasahara 1966) without citation, and simplify the expression for the total time derivative of pressure $\frac{dp}{dt}$ by omitting the horizontal pressure advection:

$$p = g \int_z \rho \, dz \tag{1}$$

$$\frac{dp}{dt} = \frac{\partial p}{\partial t} + \vec{v} \nabla p$$

$$\frac{dp}{dt} = \quad g \int_z \frac{\partial \rho}{\partial t} dz+ \qquad \vec{v}_{hor} \nabla_{hor} p + v_z \nabla_z p$$

$$\frac{dp}{dt} = \quad g \int_z [-div(\rho\vec{v})] dz+ \qquad \vec{v}_{hor} \nabla_{hor} p - v_z \rho g$$

$$\frac{dp}{dt} = \quad g \int_z [-div_{hor}(\rho\vec{v}_{hor})] dz + v_z \rho g+ \quad \vec{v}_{hor} \nabla_{hor} p - v_z \rho g$$

$$\frac{dp}{dt} = \quad g \int_z [-div_{hor}(\rho\vec{v}_{hor})] dz+ \qquad \vec{v}_{hor} \nabla_{hor} p \tag{2}$$

After that the Authors neglect the second term in the last expression above, based on the very strange scale analysis:

1. They estimate the scale of the divergence of first term by the scale of only one component. But the divergence itself is by order(s) of magnitude less than scale its components.

2. They estimate the $\nabla_{hor} p$ *dynamically* as $U^2/L$. This estimation could be correct if for example we had initially a layer at the state of rest, and then the large horizontal

scale perturbation of pressure would produce the waves (propagating with about the speed of sound). But this estimation is absolutely incorrect for weather prediction in the real atmosphere which is set in motion after cyclogenesis with zonal winds, motions of air masses and weather fronts. Indeed, the pressure advection can be sometimes very small, as well as the divergence, or sometimes the total derivative of pressure is small. But for a general case the neglect of pressure advection in favor of the divergence of mass flux may result in accumulation of additional vertical velocity.

The pressure advection is very cheap from the computations point of view, if we know the pressure and velocity. But if there was a meaningful reason to get rid of it, the much less ambiguous way would be just to further expand the last expression (2) above:

$$\frac{dp}{dt} = g \int_z [-div_{hor}(\rho \vec{v}_{hor})]dz \qquad + \qquad g\vec{v}_{hor} \int_z \nabla_{hor}\rho dz \qquad (3)$$

$$\frac{dp}{dt} = -g \int_z \rho \, div_{hor}(\vec{v}_{hor})dz \quad -g \int_z \vec{v}_{hor}\nabla_{hor}\rho dz + \quad g\vec{v}_{hor} \int_z \nabla_{hor}\rho dz \qquad (4)$$

The 2-nd and 3-d term compensate each other if velocity does not depend on $z$. So this expression

$$\frac{dp}{dt} = -g \int_z \rho \, div_{hor}(\vec{v}_{hor})dz \qquad (5)$$

is exact if $v_{hor}$ does not depend on $z$, for such a case it expresses an obvious fact that

$$\frac{dp}{dt} = -p \, div_{hor}(\vec{v}_{hor}),$$

when the pressure is the height of the air column. And it can be a fair approximation, *in contrast to omitting the pressure advection, which is not actually an approximation but an unbalancing of the expression for the total derivative of pressure.*

I will illustrate it with the motion of a wagon with sand (see the attached picture) which corresponds to the horizontal bulk transport of the masses of air. Of course you will never see such a pure motion in the real atmosphere but here I separate it to illustrate the formula.

The pressure corresponds to the height of the sand, and obviously, when the wagon moves as a whole $\frac{dp}{dt} = 0$ for any column of sand. So in the expression of the total derivative $\frac{dp}{dt} = \frac{\partial p}{\partial t} + v\frac{\partial p}{\partial x} = 0$ and $\frac{\partial p}{\partial t} = -v\frac{\partial p}{\partial x}$.

Now if we "neglect" the pressure advection $v\frac{\partial p}{\partial x}$ we will get some finite, and may be very big value of $\frac{dp}{dt}$, instead of ZERO.

Returning to the atmosphere, ***this is why the additional vertical velocity may be accumulated***, and ***this is why the energy conservation is violated***, since $\frac{dp}{dt}$ is a part of the expression for the vertical velocity and change of the full energy.

I hope that the Authors will answer this detailed claim for the mistakes, or finally retract all these mathematical inconsistencies.

———————————————————

[Figure]

$$\frac{dP}{dt} = 0 \quad ; \quad \frac{\partial P}{\partial t} = -v\frac{\partial P}{\partial x}$$

**Fig. 1.**

---

## Short Comment (SC8) · 18 Sep 2020

Dear Ilias Sibgatullin,

Thank you for your consistent attention to our paper. In some sense, all models are not correct. We have to admit that there is a controversial moment for the hydrostatic equations, in which the vertical velocity is ignored in the vertical momentum equation and evaluated at the same time. Such approximation is of course not correct for any "strange" scales, it is just an approximation for some particular scales.

Although we have answered your questions in a lot of conferences, we repeat it here for

[Discussion paper]

[Figure]

the new readers of this paper and for the improvement of the manuscript, concerning the "another two incorrectness" (the comments are in italic, and the answers are in regular letters):

1. *They estimate the scale of the divergence of the first term by the scale of only one component. But the divergence itself is by order(s) of magnitude less than scale its components.*

   **Answer:** Let's express the two terms of the right side of equation (2) in your comment as following

   $A = -g \int_z^H div\,(\rho \overrightarrow{v}_{hor}) dz',\ \ B = \overrightarrow{v}_{hor} \nabla_{hor} p.$

   From the momentum conservation equations for horizontal motion the following estimation takes place

   $$B = v_x \frac{\partial p}{\partial x} + v_y \frac{\partial p}{\partial y} = O\left( V_{hor} \times \rho \left( \frac{V_{hor}^2}{L_{hor}} + \frac{V_{hor}^2}{L_{cor}} \right) \right)$$

   Then we have the following estimations

   $$\frac{B}{\gamma p} = O\left( V_{hor} \times \frac{\rho}{\gamma p} \left( \frac{V_{hor}^2}{L_{hor}} + \frac{V_{hor}^2}{L_{cor}} \right) \right),$$

   $$\frac{A}{\gamma p} = -\frac{g}{\gamma p} \int_z^H div\,(\rho \overrightarrow{v}_{hor}) dz' = O\left( \frac{V_{hor}}{L_{hor}} \times \frac{g \widehat{\rho}\,(H-z)}{p} \right) = O\left( \frac{V_{hor}}{L_{hor}} \right)$$

Finally, using the two expressions above we get

$$\frac{B}{A} = O\left(\frac{L_{hor}}{V_{hor}}V_{hor} \times \frac{\rho}{\gamma p}\left(\frac{V_{hor}^2}{L_{hor}} + \frac{V_{hor}^2}{L_{cor}}\right)\right) = O\left(\frac{V_{hor}^2}{C^2}\left(1 + \frac{L_{hor}}{L_{cor}}\right)\right) = O\left(\mathbf{M}^2\left(1 + \frac{L_{hor}}{L_{cor}}\right)\right),$$

$$\mathbf{M} = \frac{V_{hor}}{C}, \quad C = \left(\frac{\gamma p}{\rho}\right)^{1/2} \sim 300\ m/s$$

As in this case the Mach number $\mathbf{M}$ is small, we can confidently ignore *B* with the existence of *A* in equation (2.13) of the manuscript.

2. *They estimate the $\nabla_{hor}p$ dynamically as $U^2/L$.*

**Answer:** This statement does not match the description in the manuscript. As we have mentioned above, the value of *B* ($B = \overrightarrow{v}_{hor}\nabla_{hor}p$) is evaluated by the horizontal momentum equations (1.2) and (1.3) in the manuscript. By the way, we have answered this in the reply to your previous comment:

$\frac{\partial p}{\partial x}, \frac{\partial p}{\partial y} = \rho O\left(\frac{V_{hor}^2}{L_{hor}} + \frac{V_{hor}^2}{L_{cor}}\right),$

and the neglection of *B* over *A* takes place only in equation (2.13).

We admit that in some situations (such as extreme weather, hurricanes) when both the horizontal wind and the horizontal gradient of pressure are large, the value of $B = \overrightarrow{v}_{hor}\nabla_{hor}p$ cannot be neglected.

Thank you also for your brilliant illustration with beautiful pictures, which is irrelevant to the topic of this paper. And we have to point out that in the stationary state when your wagon moves with a constant velocity $v = const$, the pressure gradient always equals to zero ($\partial p/\partial x = 0$).

[Figure]

About the violation of energy conservation, please refer to our reply to Anonymous Referee #1.

Best regards,

Robert Nigmatulin and Xiulin Xu

---

## Short Comment (SC9) · 19 Sep 2020

Dear Anonymous Referee #1,

I was referred to your review by the Authors (of the Paper under consideration). And the first thing I've noticed in your review is the phrase "In general the math appears to be sound, although as the derivations are not extensive. I have put some faith in the authors that their work is correct". Sounds nice, but it is a paper about quite unusual mathematical statements and conclusions. I've been dealing with this work (the first part of it was already published, see the comments https://arxiv.org/pdf/2001.08637.pdf) for

more than 3 years, since it directly affects my work. So I will try to most explicitly point your attention to what you are putting some faith. The mathematics here is quite simplistic and does not require sophisticated manipulations. But still I can't get the answers from the authors. Also I'm afraid you are using a bit different language from that of the Authors when you refer to the models, used by UK MetOffice. And that your concept of quasi-hydrostatic and shallow atmosphere approximation is quite more specific (according to the modern classifications) and differs from quasi-hydrostatic approximation as understood in the Paper, in which the equation of the vertical momentum is replaced just by hydrostatic balance $\dfrac{\partial p}{\partial z} = -\rho g$. This is another reason to look at the maths before comparing it to other models, just to understand **what** you are going to compare to **what**.

The authors do not give answers to the questions below, as you can see at the other branch of the discussion of this paper. So if you trust the maths of the Authors, may be you can help in answering the questions or participate in the discussion.

**1  First, you are going to trust**

is the theorem, which states that as the vertical acceleration approaches zero, the hydrostatic approximation (which admits the finite vertical velocity) is asymptotically exact. A proof of this controversial statement is not given in the paper. This statement is responsible for the strange scales, for which hydrostatic approximation is applied in the paper (from 1 km in horizontal direction) in full accordance with the Theorem.

**2 Second thing, you are going to trust**

are two incorrectness (in my humble opinion), due to which the system of equations is also incorrect for weather prediction at *any* scales.

The Authors take the traditional L.F. Richardson's framework without citation (Kasahara 1966, the minor issues of neglecting of spherical geometry, and the use of the true temperature instead of the potential/virtual temperature I even do not consider now). And after that, they simplify the expression for the total time derivative of pressure $\dfrac{dp}{dt}$ by omitting the horizontal pressure advection. Let's look at the procedure. First let us substitute the expression of hydrostatic pressure to the total derivative of pressure. We get the expressions which can be traced back to at least Richardson (1922):

$$p = g \int_z \rho \, dz \tag{1}$$

$$\frac{dp}{dt} = \frac{\partial p}{\partial t} + \vec{v} \nabla p$$

$$\frac{dp}{dt} = g \int_z \frac{\partial \rho}{\partial t} dz + \qquad \vec{v}_{hor} \nabla_{hor} p + v_z \nabla_z p$$

$$\frac{dp}{dt} = g \int_z [-div(\rho \vec{v})] dz + \qquad \vec{v}_{hor} \nabla_{hor} p - v_z \rho g$$

$$\frac{dp}{dt} = g \int_z [-div_{hor}(\rho \vec{v}_{hor})] dz + v_z \rho g + \vec{v}_{hor} \nabla_{hor} p - v_z \rho g$$

$$\frac{dp}{dt} = g \int_z [-div_{hor}(\rho \vec{v}_{hor})] dz + \qquad \vec{v}_{hor} \nabla_{hor} p \tag{2}$$

After that, the Authors neglect the second term in the last expression above, based on the very strange scale analysis:

1. They estimate the scale of the divergence of first term by the scale of only one component. But the divergence itself is by order(s) of magnitude less than scale its components.

2. They estimate the $\nabla_{hor}p$ *dynamically* as $\rho\, U^2/L$. This estimation could be correct if for example we had initially a layer at the state of rest, and then the large horizontal scale perturbation of pressure would produce the waves (propagating with about the speed of sound). But this estimation is absolutely incorrect for weather prediction in the real atmosphere which is set in motion after cyclogenesis with zonal winds, motions of air masses and weather fronts. Indeed, the pressure advection can be sometimes very small, as well as the divergence, or sometimes the total derivative of pressure is small. But for a general case the neglect of pressure advection in favor of the divergence of mass flux may result in accumulation of additional vertical velocity.

The pressure advection is very cheap from the computations point of view, if we know the pressure and velocity. But if there was a meaningful reason to get rid of it, the much less ambiguous way would be just to further expand the last expression (2) above:

$$\frac{dp}{dt} = g \int_z [-div_{hor}(\rho\vec{v}_{hor})]dz \qquad\qquad + \qquad\qquad g\vec{v}_{hor}\int_z \nabla_{hor}\rho\, dz \qquad (3)$$

$$\frac{dp}{dt} = -g \int_z \rho\, div_{hor}(\vec{v}_{hor})dz \quad -g\int_z \vec{v}_{hor}\nabla_{hor}\rho\, dz+ \quad g\vec{v}_{hor}\int_z \nabla_{hor}\rho\, dz \qquad (4)$$

The 2-nd and 3-d term compensate each other if velocity does not depend on $z$. So this expression

$$\frac{dp}{dt} = -g \int_z \rho\, div_{hor}(\vec{v}_{hor})dz \qquad (5)$$
is exact if $v_{hor}$ does not depend on $z$, for such a case it expresses an obvious fact that

$$\frac{dp}{dt} = -p\,div_{hor}(\vec{v}_{hor}),$$

when the pressure is the height of the air column. And it can be a fair approximation, ***in contrast to omitting the pressure advection, which is not actually an approximation but an*** **unbalancing** *of the expression for the total derivative of pressure.*

I will illustrate it with the motion of a wagon with sand (see the attached picture) which corresponds to the horizontal bulk transport of the masses of air. Of course, you will *never see such a pure motion* in the real atmosphere, since it may be a king of seconary flow compared to global waves. But here I put it separately just to illustrate the formula.

The pressure corresponds to the height of the sand, and obviously, when the wagon moves as a whole $\frac{dp}{dt} = 0$ for any column of sand. So in the expression of the total derivative $\frac{dp}{dt} = \frac{\partial p}{\partial t} + v\frac{\partial p}{\partial x} = 0$ and $\frac{\partial p}{\partial t} = -v\frac{\partial p}{\partial x}$.

Now if we "neglect" the pressure advection $v\frac{\partial p}{\partial x}$, we will get some finite, and may be very big value of $\frac{dp}{dt}$, instead of ZERO.

Returning to the atmosphere, ***this is why the additional vertical velocity may be accumulated***, and ***this is why the energy conservation is violated***, since $\frac{dp}{dt}$ is a part of the expressions for the vertical velocity and the change of the full energy.

**3 The last thing you are going to trust is the approach to prepare the system for linear analysis.**

The hydrostatic approximation is not an evolutionary system, since one of the equations is diagnostic, not prognostic. The traditional way to make it evolutionary is the discretizations of the vertical operator. There are numerous works devoted to stability and correctness of the hydrostatic approximation, which had proved that discretizations used in the major weather prediction systems are well posed.

Instead of this rich experience the Authors declare a new variable "$\dot{M}$" as independent of $M$ and make linear analysis of this bigger system of equations for functionally dependant variables. Besides the fact that such an introduction makes the system degenerate, it becomes difficult to make comparison with the well known results. So my question is what are the benefits of such "evolutionarization" as compared to traditional way, and why not making the analysis the usual comprehensible way, compare with the previous results, and underline the new idea.

[Figure]

[Figure]

**Fig. 1.**

---

## Short Comment (SC10) · 20 Sep 2020

Dear Authors,

> *Thank you for your consistent attention to our paper.*

Not at all, since your theorem directly contradicts the existence of convection and gravity waves (except for the long ones), you can count on my constant attention further on.

> *Although we have answered your questions in a lot of conferences.*

[Figure]

Actually you (the first Author) did not. I did not get the answers to my questions. Instead, I am being fired by the first Author. And I will show below, that although now you have put some formulas, you reply just does not contain anywhere the answers to my direct questions.

**1 The main question**

Even now you did not answer the question about the proof of the statement in your theorem, that for small vertical acceleration (normalized by gravity), the hydrostatic approximation (with equation for change of vertical momentum being replaced by hydrostatic balance) is asymptotically exact. *Please, give a proof, or otherwise can you tell that it is your secret know-how?*

This is the most important question I ask you for more than 3 years, since I consider such a theorem to be very harmful in the walls of a University. All other mistakes are of minor importance, because nowadays nobody uses L.F. Richardson's framework for weather prediction, and it is important mostly for historical reasons.

**2 About your 1-st answer on the question *1. They estimate the scale of the divergence of first term by the scale of only one component. But the divergence itself is by order(s) of magnitude less than scale its components.**

Again in your "answer" you continue to estimate $div_{hor}(\rho\vec{v}_{hor})$ as $\hat{\rho}\,\dfrac{V_{hor}}{L_{hor}}$, so as a scale of one component without any explanation. Is it so? Do not put other formulas, just answer this question, that I have asked. I repeat again that this estimation is not correct for large-scale atmospheric dynamics.

You begin $B, \gamma, M$ and other quantities for estimation of pressure advection, but it was not a part of the question.

So, to count down this question as answered, please answer directly only about the estimation of the divergence of the horizontal mass flux, that is $\mathrm{div}_{hor}(\rho\vec{v}_{hor})$.

**3 About your 2-d answer on the question *2. They estimate the* $\nabla_{hor}p$ **dynamically *as* $\rho U^2/L$.**

I've made a misprint by omitting $\rho$, sorry for that. Yes, I've meant exactly what you put in your "answers" again and again, that $\dfrac{\partial p}{\partial x}, \dfrac{\partial p}{\partial y} = \rho\, O\left(\dfrac{V_{hor}^2}{L_{hor}} + \dfrac{V_{hor}^2}{L_{hor}}\right)$. For a general case of large-scale atmospheric dynamics this estimation is wrong for the reasons, I've mentioned in my previous message.

Do you want one of the counter-examples? Take the geostrophic balance

$$f \cdot v = \ \frac{1}{\rho}\frac{\partial p}{\partial x} f \cdot u = -\frac{1}{\rho}\frac{\partial p}{\partial y}$$

Here $\dfrac{\partial p}{\partial x} = \rho\, f\, O\left(V_{hor}\right)$. But not only that what is important. The atmosphere accumulates the mechanical energy, so the velocity in the zonal winds and motion of the air masses can not always be explained straightforwardly just due to the pressure gradients, as if it had appeared from a state of rest.

[Figure]

**4**  $>$ *About the violation of energy conservation, please refer to our reply to Anonymous Referee* $\#1$*.*

Your reply to Anonymous Referee $\#1$ does not contain the answer the problem of the conservation of the full energy, if $\dfrac{dp}{dt}$ is not correctly computed.

**5**  **Finally,** *And we have to point out that in the stationary state when your wagon moves with a constant velocity v = const, the pressure gradient always equals to zero* $\left(\frac{\partial p}{\partial x} = 0\right)$*.*

I would recommend to read some books on mathematical physics about the concepts of total and partial derivative.

In the picture I attach again the pressure somewhere on the bottom of the wagon is roughly equal to height of the column of sand above it (it works better if the wagon is very long). I've tried to show that it is the total derivative of pressure that is equal to zero, since the form of the surface of sand does not change with respect to the wagon.

But the partial derivative of the pressure over $x$ is not equal to zero, since the level of sand near the borders is much less than in the center.

I'm sorry, that you can not get the idea even from the such an illustration.

To make the illustration more vivid, imagine an action movie, and that somebody tries to escape a prison in a wagon with sand. If you hide below the sand on the bottom near the right border, the weight of the sand over you will be bearable, and there are chances that you will survive. But if a humankind will hide at the bottom at the center of the wagon, he can be smashed by the weight of the tons of sand above him. So the pressure is different along the wagon, and $\dfrac{\partial p}{\partial x}\Delta x$ is the drop of pressure over $\Delta x$.

Now, if the wagon will move along the railway, but you will stay on the bottom without any motion with respect to the wagon, the total derivative of pressure $\frac{dp}{dt}$ will be exactly ZERO. Now we remember that $\frac{dp}{dt} = \frac{\partial p}{\partial t} + v\frac{\partial p}{\partial x} = 0$, $\frac{\partial p}{\partial t} = -v\frac{\partial p}{\partial x}$, so partial derivative of pressure is equal to its advection, and can take quite a big value. So if you now "neglect" advection of pressure $v\frac{\partial p}{\partial x}$ you will get some strange and may be quite a big value of $\frac{dp}{dt}$ instead of ZERO.
* * *
[Figure]

**Fig. 1.**

[Figure]

---

## Short Comment (SC11) · 20 Sep 2020

Dear Ilias Sibgatullin,

Thank you again for the comments. And sorry to the readers for such tedious discussions. Some clarifications are made for a better understanding of mathematics in the manuscript. The numbering corresponds to that of the previous comment (SC10).

1. The reason why the quasi-hydrostatic equation can be used in the long-term climatic modeling is that under such scales, in the vertical momentum equation the vertical acceleration term is negligibly small in comparison with gravity. However, the quasi-

hydrostatic equation is not valid for small scale processes and extreme weather, where the vertical motion is essential. The theorem is proved under the condition that the inertial forces are negligibly small in comparison with gravity.

2. According to the previous answer (SC8), the term A is estimated using the Mean Value Theorem for Integral. If we focus on one altitude, the terms of divergence can sometimes compensate, but our estimation is made for the integral over all levels of the vertical column.

3. For this question please refer to SC3 (eq 1) and SC8 (answer #1). Your example of geostrophic balance is not a counterexample; it is identical to eq(1) of SC3 if the acceleration term is not neglected in the momentum equation.

4. We stick to the answer in AC4.

5. "But the partial derivative of the pressure over x is not equal to zero, since the level of sand near the borders is much less than in the center." As far as I can understand, the partial derivative of pressure over x is calculated at the same level.

I am sorry that our answers do not satisfy you. We will of course try our best to improve the manuscripts to make the derivations more clear for the readers. We are very happy to hear from you again if you have any new doubts or questions associated with the content of this manuscript. But please do not repeat the same questions, or the readers can easily get confused. I hope other readers can make some comments about this discussion. Thank you again for your time.

Best regards,

Xiulin Xu
* * *

---

## Short Comment (SC12) · 20 Sep 2020

In the context of my original comment, this is an almost predictable back-and-forth discussion. As I implied, in the absence of some actual data to compare against, the mathematical degrees of freedom involved in a fluid dynamics formulation will allow many possibilities for a potential solution.

Geoscientific model development should be geared toward emulating some known physical behavior, otherwise it would be just discussion of the math underlying partial differential equations. Consider the figure below taken from https://doi.org/10.1029/2019JD032362 which collects over 30 model comparisons to a

historical QBO time-series, yet revealing little consilience and limited agreement among the models. The authors of the paper suggest that even though "the number of climate or Earth system models being able to simulate the QBO" .... "However, the quality of the simulation of the QBO has not improved."

What can end this back-and-forth discussion is the mathematical insight leading to a formulation that will allow the QBO time-series to be modeled effectively. Otherwise, in the absence of a real-world context, there is no end in sight.

1. Richter, J. H., Anstey, J. A., Butchart, N., Kawatani, Y., Meehl, G. A., Osprey, S., & Simpson, I. R. (2020). Progress in simulating the quasi‐biennial oscillation in CMIP models. Journal Geophysical Research: Atmospheres

———————————————

[Figure]

**Fig. 1.**

---

## Referee Comment (RC2) · Anonymous Referee #2 · 28 Sep 2020

This paper tries to analyse wavelike perturbations from a quasi-hydrostatic equation set. This equation set is put into a framework in z-coordinates. This lets the equations look like as they are formulated as we are used to them in p-coordinates (hence uses some kind of the Richardson equation). The paper seeks for potential unstable growing solutions and finds them. However, in my opinion, the results are questionable, because they are not leading to the results which are usually obtained, namely the solution of gravity waves and their dispersion relation. And as we know, GW solutions are not growing in time, but are pure waves.

My general impression is that since the underlying physics of the system does not

differ from other analyses to be be found in the literature, the same known results should come to the fore. A dispersion analysis of waves in a system should not depend on the specific prognosic variables or differently written equations. All those systems must lead to the same results, but equation (4.3) does not coincide with other known solutions.

I tried to figure out where the problem might be in the actual derivation. Two points are somehow strange to me. First, the system (2.23) consists of 6 instead of 5 equations. So, there are some linear dependencies among the equations. But, this might be not so problematic, since the found dispersion relations finds an equation in $\omega^3$, which means three solutions, which is fine. Second, compared to more traditional approaches, the step of the Bretherton (1966) transfromation has not been done. Consequently the coefficient matrices do not have constant coefficients (appendix A), because the density depends on height. I do not know, which consequences arise due to this missing step.

In the following, I copy the part of my lecture of gravity waves, which focuses on the derivation of them under the hydrostatic constraint (omission of blue terms) and under shedding of acoustic waves (omission of red terms). Perhaps the authors could figure out, how their derivation differs from these conventional steps and how their approach could be brought under the umbrella of known results.

Here follows the lecture part:

**1   Gravity waves (GWs)**

**1.1   Dispersion relation for gravity waves**

- background state assumes arbitrarily constant $N^2$ (includes the special isothermal case $N_{iso}^2 = g^2/(c_p T)$)

[Figure]

- hydrostatic approximation is not needed

- assuming incompressibility is not needed, acoustic waves are later separated in the dispersion relation itself

- it becomes obvious that the amplitudes of the wave perturbations have exponential behavior with height

- a constant Coriolis parameter $f$ is assumed

- for simplicity we assume a dry atmosphere

- irreversible processes like friction or heating are not included in the wave analysis (later we will include them)

The governing equations are linearized around a hydrostatic state and a mean zonal current $U$, and the individual time derivative is abbreviated with $d_t \cdot = \partial_t \cdot + U \partial_x \cdot$. The vertical advection is separated out from this operator. We have

$$u = U + u', \qquad v = v', \qquad w = w', \qquad p = p_0(z) + p', \qquad \theta = \theta_0(z) + \theta' \quad (1)$$

The equation of state is assumed for the mean state / the background separately: $p_0(z) = \varrho_0(z) R T_0$. The hydrostatic relation hold for the background state $\partial p_0 / \partial z = -\varrho_0(z) g$. And we have $T_0(z) = \theta_0(z) \Pi_0(z)$.

As the thermodynamic equation we use the potential temperature equation which reads

$$d_t \theta' + w' \partial_z \theta_0(z) \;=\; 0 \qquad | \cdot g/\theta_0(z) \quad (2)$$
$$d_t b' + w' N^2 \;=\; 0 \quad (3)$$

where $b' = g\theta'/\theta$ is called the buoyancy. Brunt-Vaisala frequency / buoyancy frequency: $N^2 = g \partial_z \ln \theta_0(z)$.

The continuity equation is not used directly, but rather, an equation for the pressure perturbation is chosen as prognostic variable. Then we have as the not yet linearized equation

$$\varrho c_v d_t T + p \nabla \cdot \mathbf{v} = 0 \tag{4}$$

$$\varrho c_v d_t \frac{p}{R\varrho} + p \nabla \cdot \mathbf{v} = 0 \tag{5}$$

$$\frac{\varrho c_v}{R\varrho} d_t p - \varrho c_v \frac{p}{R\varrho^2} d_t \varrho + p \nabla \cdot \mathbf{v} = 0 \tag{6}$$

$$\frac{c_v}{R} d_t p + \frac{\varrho^2 p c_v}{R\varrho^2} \nabla \cdot \mathbf{v} + p \nabla \cdot \mathbf{v} = 0 \tag{7}$$

$$\frac{c_v}{R} d_t p + \frac{(c_v + R)p}{R} \nabla \cdot \mathbf{v} = 0 \tag{8}$$

$$\frac{c_v}{c_p R T} \frac{1}{\varrho} d_t p + \nabla \cdot \mathbf{v} = 0 \tag{9}$$

$$\frac{1}{c_s^2 \varrho} d_t p + \nabla \cdot \mathbf{v} = 0 \tag{10}$$

Speed of sound: $c_s^2 = c_p R T / c_v$. Note that the vertical advection of $p_0(z)$ remains relevant when linearizing.

Original linearized equations are

$$d_t u' - f v' + \frac{1}{\varrho_0(z)} \partial_x p' = 0 \tag{11}$$

$$d_t v' + f u' + \frac{1}{\varrho_0(z)} \partial_y p' = 0 \tag{12}$$

$$d_t w' - b' + \frac{g}{c_s^2} \frac{p'}{\varrho_0(z)} + \frac{1}{\varrho_0(z)} \partial_z p' = 0 \tag{13}$$

$$d_t b' + w' N^2 = 0 \tag{14}$$

$$\frac{1}{c_s^2 \varrho_0(z)} d_t p' - \frac{g}{c_s^2} w' + \nabla \cdot \mathbf{v}' = 0 \tag{15}$$

The blue term vanishes for the hydrostatic constraint. The red term in (15) vanishes for incompressibility. In (13) the red term vanishes if the pseudo-density $\varrho^* = \rho_0(z)\theta_0(z)/\theta$ is used in the pressure gradient term. These conditions help filtering out sound waves. We must now erase the height dependency of the density. Key tool is the Bretherton transformation[1]. Define transformation: $(u'', v'', w'', b'') = \sqrt{\varrho_0(z)/\varrho_{surf}}(u', v', w', b')$ and $p'' = \sqrt{\varrho_{surf}/\varrho_0(z)} \; p'$. This transformation guarantees the exponential increase of wave amplitudes with height due to the density decrease (see earlier chapter on Rossby waves).

After transformation the system becomes

$$d_t u'' - f v'' + \frac{\partial_x p''}{\varrho_{surf}} = 0 \tag{16}$$

$$d_t v'' + f u'' + \frac{\partial_y p''}{\varrho_{surf}} = 0 \tag{17}$$

$$d_t w'' - b'' + \left(\frac{g}{c_s^2} - \frac{1}{2H}\right) \frac{p''}{\varrho_{surf}} + \frac{\partial_z p''}{\varrho_{surf}} = 0 \tag{18}$$

$$d_t b'' + w'' N^2 = 0 \tag{19}$$

$$\frac{d_t p''}{c_s^2 \varrho_{surf}} + \nabla_h \cdot \mathbf{v}_h'' + \left(\frac{1}{2H} - \frac{g}{c_s^2}\right) w'' + \partial_z w'' = 0 \tag{20}$$

Scale height:

$$\frac{1}{H} = -\frac{1}{\varrho_0} \frac{\partial \varrho_0}{\partial z} = \frac{N^2}{g} + \frac{g}{c_s^2} \tag{21}$$
* * *
[1]Bretherton FP. 1966. The propagation of groups of internal gravity waves in a shear flow. QJRMS 92: 466-480.

Wave ansatz for an arbitrary transformed variable: $\psi'' = A_\psi \exp(i(kx + ly + mz - \omega t))$

Intrinsic frequency: $\omega_I = \omega - kU$

Linear equation system:

$$
\begin{pmatrix}
-i\omega_I & -f & 0 & 0 & ik \\
f & -i\omega_I & 0 & 0 & il \\
0 & 0 & -i\omega_I & -1 & im - \frac{1}{2H} + \frac{g}{c_s^2} \\
0 & 0 & N^2 & -i\omega_I & 0 \\
ik & il & im + \frac{1}{2H} - \frac{g}{c_s^2} & 0 & \frac{-i\omega_I}{c_s^2}
\end{pmatrix}
\cdot
\begin{pmatrix}
A_u \\
A_v \\
A_w \\
A_b \\
A_{p/\varrho_{surf}}
\end{pmatrix}
= 0
\qquad (22)
$$

det (...) = 0 defines the dispersion relation

$$
\begin{aligned}
\det(...) &= -i\omega_I \left\{ -i\omega_I \left( i\omega_I\omega_I \frac{\omega_I}{c_s^2} + i\omega_I \left( im + \frac{1}{2H} - \frac{g}{c_s^2} \right) \left( im - \frac{1}{2H} + \frac{g}{c_s^2} \right) - \frac{i\omega_I}{c_s^2} N^2 \right) \right. \\
&\qquad \left. -il \left( -il\omega_I\omega_I + ilN^2 \right) \right\} \\
&\quad + f \left\{ f \left( i\omega_I\omega_I \frac{\omega_I}{c_s^2} + i\omega_I \left( im + \frac{1}{2H} - \frac{g}{c_s^2} \right) \left( im - \frac{1}{2H} + \frac{g}{c_s^2} \right) - \frac{i\omega_I}{c_s^2} N^2 \right) \right. \\
&\qquad \left. -il \left( -ik\omega_I\omega_I + ikN^2 \right) \right\} \\
&\quad + ik \left\{ f \left( -il\omega_I\omega_I + ilN^2 \right) + i\omega_I(-ik\omega_I\omega_I + ikN^2) \right\} \\
&= 0
\end{aligned}
\qquad (23)
$$

This gives

$$
\begin{aligned}
0 &= (f^2 - \omega_I^2) \left( i\omega_I\omega_I \frac{\omega_I}{c_s^2} + i\omega_I(im + \frac{1}{2H} - \frac{g}{c_s^2})(im - \frac{1}{2H} + \frac{g}{c_s^2}) - \frac{i\omega_I}{c_s^2} N^2 \right) \\
&\quad -il^2\omega_I(-\omega_I\omega_I + N^2) + lkf(-\omega_I\omega_I + N^2) - klf(-\omega_I\omega_I + N^2) - ik^2\omega_I(-\omega_I\omega_I + N^2)
\end{aligned}
\qquad (24)
$$

And shorter

$$
0 = [f^2 - \omega_I^2] \left[ \frac{\omega_I}{c_s^2}(\omega_I\omega_I - N^2) + \omega_I \left[ -m^2 - \left[ \frac{1}{2H} - \frac{g}{c_s^2} \right]^2 \right] \right] + \omega_I(l^2 + k^2)(\omega_I\omega_I - N^2)
\qquad (25)
$$

$$0 = \frac{\omega_I}{c_s^2}(\omega_I\omega_I - N^2)(f^2 - \omega_I^2) + \omega_I\left[(\omega_I^2 - f^2)\left[m^2 + \left[\frac{1}{2H} - \frac{g}{c_s^2}\right]^2\right] + (l^2 + k^2)(\omega_I\omega_I - N^2)\right] \quad (26)$$

The red terms are significant for acoustic waves. For $g = 0$, $N^2 = 0$ and $f^2 = 0$ holds: $\omega_{I,ac}^2 = c_s^2(k^2 + l^2 + m^2)$.

A stationary solution, the Rossby mode, $\omega_I = 0$, exists in any case, because $\beta = 0$.

GWs are derived by neglecting all the red terms, hence acoustic waves are filtered out. Hydrostatic GWs disregard the blue term. This gives

$$\omega_I^2 = f^2 + \frac{N^2(k^2 + l^2)}{m^2 + \frac{1}{4H^2}} \quad (27)$$

General GWs have the dispersion relation

$$\omega_I^2 = \frac{f^2(m^2 + \frac{1}{4H^2}) + N^2(k^2 + l^2)}{k^2 + l^2 + m^2 + \frac{1}{4H^2}} = \left(f^2 + \frac{N^2(k^2 + l^2)}{m^2 + \frac{1}{4H^2}}\right)\frac{m^2 + \frac{1}{4H^2}}{k^2 + l^2 + m^2 + \frac{1}{4H^2}} \quad (28)$$

---

## Short Comment (SC13) · 2 Oct 2020

Dear Sir or Madam:

Thank you very much for reviewing our manuscript. We will try our best to clarify questionable moments and figure out the differences between our set of equations and yours in the following text.

The dispersion relation in the manuscript differs from that in your GW analysis due to the following differences:

1. For the horizontal momentum equations, the term $U\partial_z$ is ignored in unlinearized

equations of (11) and (12). While in our set of equations, we use the complete horizontal momentum equations, the second and third equations of (2.23).

2. For the vertical momentum equation, the buoyancy term $-b'$ is included in (13). In our set of equations, the vertical inertial is totally ignored in the vertical momentum equation, the sixth equation of (2.23).

3. The continuity equation uses the pressure as a prognostic variable in the unlinearized equation of (15). In our equations, the complete continuity equation with density as a prognostic variable is used, the first equation of (2.23).

4. The thermodynamic equation is explicitly used in your set of equations, the unlinearized equation of (14). Instead of this, we use the thermodynamic equation to evaluate the vertical velocity and use the fourth equation of (2.23) in the set of equations.

5. You have five equations and five independent variables. To close the system of equations, we use a new independent variable $\dot{M}$ and the equation corresponding to this variable, the fifth equation of (2.23).

Besides, in our paper the frequency $\overline{\omega}_*$ is complex, the positive sign of the imaginary part $\overline{\omega}_{**}$ corresponds to exponential growth of perturbation, i.e., instability. The purpose of the linear analysis of the system of equations (2.23) is to find out situations when perturbations (usually caused by numerical solutions to the original system of equations) grow or decay. The fact that "*The coefficient matrices do not have constant coefficients (appendix A)*" do not affect the result of dispersion relation, it allows us to study the instability property of different basic solutions to the original system, including the basic solution in your GW analysis (a hydrostatic state and a mean zonal current $U$).

Moreover, from the dispersion relation (27) we can get a result similar to (4.8) in the manuscript if we consider the frequency $\omega_I$ as complex. When $N^2 < 0$ and

$k^2 + l^2 \gg m^2$, we can get a root of $\omega_I$ with positive imaginary part, in particular, when $\left(k^2 + l^2\right)/m^2 \to \infty$, the solution is absolute unstable by definition (3.14). In the situation of Gravity Waves in a stably stratified atmosphere $N^2 > 0$, the dispersion relation (27) only gives a real root of $\omega_I$, describing the intrinsic oscillation frequency of air deviating from the basic state.

Sincerely,

Xiulin Xu
* * *

---

## Short Comment (SC14) · 6 Oct 2020

I was pointed to a misprint in my previous comment

[Figure]

. . .

$$\frac{dp}{dt} = -g \int_z \rho \, div_{hor}(\vec{v}_{hor}) dz$$

is exact if $v_{hor}$ does not depend on $z$, for such a case it expresses an obvious fact that

$$\frac{dp}{dt} = -p \, div_{hor}(\vec{v}_{hor}),$$

when the pressure is the height of the air column . . .

Of course, I meant the weight of a column of air. There is a direct analogy to shallow water equations here, so I've made a misprint.

Another question was about "my" dimensional analysis of the total derivative of pressure, if I do not like the analysis by the Authors.

The analysis below is given to show that, in my opinion, primitive scale analysis of the primitive equations is not enough for definite conclusions whether the pressure advection can be dropped during the time integration of the equations. Such analysis is correct as an overall estimate, but it can not predict the importance of zonal or bulk flows. Also I've put a review at

https://pubpeer.com/publications/CDC804462F2FC7E0826F4E0E09BB4E

where the major concern is a proof of the Theorem, from which it follows, that short waves have to transform to long waves, as acceleration becomes smaller.

First, I would recall the expression for the total derivative of pressure in hydrostatic

approximation before any simplifications (3)

$$\frac{dp}{dt} = g \int_z [-div_{hor}(\rho\vec{v}_{hor})]dz + g\vec{v}_{hor} \int_z \nabla_{hor}\rho dz$$

The pressure advection is the second term, which can be estimated as

$$g\,U\,\frac{\Delta\rho}{L_{hor}}\tilde{H},$$

where $\Delta\rho$ is the change of $\rho$ over the horizontal scale, $\tilde{H} = H - z$.

The first term could be very overroughly estimated as :

$$g \int_z \frac{\partial\rho}{\partial t}dz = g\,\frac{\Delta\rho}{\tau}\tilde{H} = g\,U\,\frac{\Delta\rho}{L_{hor}}\tilde{H},$$

where $\tau$ is the time scale, so $\tau = L_{hor}/U$. Here the vertical convection is neglected in the same manner as the Authors do not account for vertical motion, when they try to estimate pressure advection as $\rho\,U^2/L_{hor}$. For such a case the orders of the first and second terms are equal, so pressure advection can not be neglected.

But let's use the most direct approach for estimation of the first term as

$$g\frac{\hat{\rho}DIV(v_{hor})}{L_{hor}}\tilde{H},$$

where $DIV(v_{hor})$ – estimation of the horizontal divergence.

Hence, the ratio of the pressure advection and the first term can be estimated as

$$\frac{U}{DIV(v_{hor})}\frac{\Delta\rho}{\rho} = \frac{\Delta\rho}{\alpha\rho},$$

where $\alpha$ is the factor by which the divergence of the velocity is less than the velocity itself. For the synoptic scale motions such a factor $\alpha$ is about $1/10$ (Charney, 1948). On the other hand, the pressure (and density) changes routinely by about $\dfrac{1}{100}$ or even up to $\dfrac{1}{20}$ over synoptic scale (see an attached example of the pressure map over Europe), so: $\dfrac{\Delta\rho}{\rho} \approx \dfrac{1}{100}$ or sometimes even $\approx \dfrac{1}{50}$.

The result so far is that by the most direct approach above, the ratio of the pressure advection to the first term in (3) for synoptic scale motions $\dfrac{\Delta\rho}{\alpha\rho}$ can be estimated by about $\dfrac{1}{10}$ or even $\dfrac{1}{5}$.

Such a ratio may be enough to neglect pressure advection in tendency equation, when one wants to roughly know the pressure for tomorrow. But for time integration of the primitive equations with small but quite important vertical velocity expressed with the help of $\dfrac{dp}{dt}$, such a one-sided deficiency may provoke the accumulation of the residual, and result in additional vertical velocity and violation of the conservation of the full energy.

So even such a direct scale analysis does not give me a definite answer whether pressure advection can be neglected for time integration of the primitive equation. Decomposition of the velocity to geostrophic and ageostrophic also does not give a definite answer, since both pressure advection and divergence vanish in purely geostrophic limit. So I'm not sure that such a conclusion is possible to formulate as a theorem. It would be interesting to find a manuscript where the ratio of the pressure advection to the divergence term is estimated for motions at different scales, and at different latitudes. Even not so much from practical point of view, as for qualitative understanding. May be indeed there are situations at some latitudes, when pressure advection can be always ignored during time-integration of primitive equations, I am ignorant about that.

*The scale analysis above does not prove, that advection of pressure is always substantial*. It shows only, that such an analysis is not enough for neglect of pressure advection in integration of the equations, and more delicate approach is needed. The approach undertaken by the Author, which is based on the *estimation of the mass flux divergence by scale of only one component, and estimation of pressure gradient as $\rho\, U^2/L$ is wrong at any scales of weather prediction problems* in hydrostatic approximation (I wonder if someone can give a counterexample, assuming the initial conditions are not special), so it can lead to *useless* models.

———————————————————

[Figure]

**Fig. 1.**

$$\frac{dv_z}{dt} \rightarrow 0$$

???

**Fig. 2.**

---

## Short Comment (SC15) · 6 Oct 2020

Of course you are right about emulating some known physical behavior, and also a citation

> "all models are wrong, but some are useful"

at the principle page of `www.geoscientific-model-development.net` seems to be very appropriate. But the Authors do not want to validate their model agaist your example or any other, they want just to declare a Theorem without showing any proof.

[Figure]

At the same time, when a result is formulated as a theorem, it can only be either *true* or *wrong*. And a wrong theorem may result in denying the *useful* models and appearance of *useless* models. I suppose that this is the case with the Paper under consideration and also with the previous Paper of the first author https://doi.org/10.1134/S0015462818040201.

This is why I am asking the Authors to give a proof of the claim in their Theorem, that from the smallness of the vertical acceleration it follows, that equation for the change of the vertical momentum can be replaced by hydrostatic balance while assuming the finite vertical acceleration. To my knowledge, the correct asymptotics should be given by the $\varepsilon = H/L$, where H and L are the vertical and horizontal scales of motion. Strictly speaking, if the Theorem in the Paper was true, the Life on Earth would not appear. At least as we know it, since restoring force about 0.01 of gravity would be always "neglected", and no typical flows with convection, internal waves (except for long ones) etc. would arise in Ocean and Atmosphere.

And it is not just a philosophical question. The first Author is actually insisting on application of his theorem in Moscow University.

I still hope, that a result of this public discussion, the Authors will retract the Theorem in this Paper, and also the first author will retract the paper https://doi.org/10.1134/S0015462818040201 where the Theorem appeared for the first time. Or otherwise, the Authors will finally present a proof of the Theorem, so it can be publicly acknowledged as true or false.

---

## Short Comment (SC16) · 7 Oct 2020

Dear Ilias Sibgatullin,

1. According to your comments and suggestions, we will change some formulations in the manuscript to make the derivations more rigorous. Changes would be make in the analysis for $\mathrm{d}p/\mathrm{d}t$. For convinence, please first refer to SC8. In SC8 and the manuscript, we drop the term $B = \overrightarrow{v}_{hor}\nabla_{hor}p$ by comparing it with $A = -g\int_z^H div\,(\rho\overrightarrow{v}_{hor})\mathrm{d}z^{'}$.

   However, you are not satisfied with the process of estimating $A$. Thus, we only

give an estimation of $B$:

$$B = v_x \frac{\partial p}{\partial x} + v_y \frac{\partial p}{\partial y} = O\left(V_{hor} \times \rho \left(\frac{V_{hor}^2}{L_{hor}} + \frac{V_{hor}^2}{L_{cor}}\right)\right)$$

$$\frac{B}{\gamma p} = O\left(V_{hor} \times \frac{\rho}{\gamma p}\left(\frac{V_{hor}^2}{L_{hor}} + \frac{V_{hor}^2}{L_{cor}}\right)\right) = O\left(\frac{\mathbf{M}^2}{\tau}\right)$$

We then drop the term $B$ due to the smallness of the Mach number $\mathbf{M}$ instead of comparing it with $A$, and only use the term $A$ to estimate $\mathrm{d}p/\mathrm{d}t$. We would also change the manuscript in this point.

2. The purpose of this paper is to analyze the shortwave instability of the quasi-hydrostatic equations.

3. Please do not make any comments on anything that has nothing to do with the manuscript's content. Malicious comments on the authors are not welcomed.

Best regards,

Xiulin Xu

---

## Short Comment (SC17) · 9 Oct 2020

*Dear Robert Nigmatulin and Xiulin Xu,*

[Figure]

**1 "For the vertical momentum equation, the buoyancy term $-b'$ is included in (13). In our set of equations, the vertical inertial is totally ignored in the vertical momentum equation, the sixth equation of (2.23)."**

Is there any connection between "vertical inertial is totally ignored" and ignoring of buoyancy term $-b'$? Do you want to tell that you have ignored buoyancy perturbations in your hydrostatic approximation model, and vertical gradient of pressure deviation is no more balanced? In such a case vertical hydrostatic balance would be violated and its not very clear what you are studying. Or may be you have misinterpreted your own writings?

**2 " To close the system of equations, we use a new independent variable $\dot{M}$ and the equation corresponding to this variable, the fifth equation of (2.23).'**

Let's look at your "independent" variables:

$$(\rho, v_x, v_y, v_z, \dot{M}, M),$$

where

$$M = \int_z \rho \, dz \tag{1}$$

$$\dot{M} = -\int_z div_{hor}(\rho v_{hor}) dz \tag{2}$$

(such a notation is strange for me since $\dot{M} \neq \dfrac{dM}{dt}$, especially in the walls of the faculty of mechanics and mathematics, but you can do it).

[Figure]

I thought before that you made such a trick to make the system evolutionary, since from above it follows f.e.

$$\frac{\partial M}{\partial t} = \dot{M} + \rho v_z \tag{3}$$

But now I've looked in the appendix A for the matrix $B_t$ and it looks like you did not even used the expression above for $\dfrac{\partial M}{\partial t}$! Instead, you put in your matrices the expression $\dfrac{\partial M}{\partial z} = -\rho$.

Only three equation in your six-equations system for perturbations are evolutionary, i.e. they have $\dfrac{\partial}{\partial t}$. And still you give the $6 \times 6$ matrix for linear analysis, as if it was for $6 \times 6$ evolutionary system. It's an amazing approach, but I am lost now what is the connection of such an approach to the analysis of perturbations of the hydrostatic approximation.

---

## Short Comment (SC18) · 9 Oct 2020

Dear Ilias Sibgatullin,

According to your two comments, we have the following answers:

1. We get the linearized equation of the quasi-hydrostatic equation directly from the sixth equation of (2.23) by giving a perturbation to the basic solution (see eq. (3.7)). Thus, different from (13) in RC2, we have no bouncy term $-b'$ in the linearized equation. However, it is not shown how (13) is obtained in RC2.

[Figure]

2. Sorry for the error of the "independent" variable. As a dependent variable in our set of equations, $\dot{M}$ is defined by (2.11). This variable also appears in the continuity equation and the equation for vertical velocity. If the denotation is strange to you, we can use another symbol for the convenience of the readers. We have only 3 diagnostic equations, and it seems to you not appropriate to conduct linear analysis even though all the derivations are mathematically strict. Please refer to RC2, not all the five equations (11)-(15) are diagnostic.

Best regards,

Xiulin Xu

---

## Short Comment (SC19) · 11 Oct 2020

The term used in SC18 "diagnostic equations" might be confusing. Here we place it more specifically: in the set of equations (2.23), there are three time-dependent (or prognostic) equations - the first three equations; and three time-independent (or diagnostic) equations - the last three equations.

Best regards, Xiulin Xu
* * *

---

## Short Comment (SC20) · 12 Oct 2020

Dear Robert Nigmatulin and Xiulin Xu,

You try to make conclusions on the well-posedness of the Cauchy problem. For evolution system it means hyperbolicity. But the system of hydrostatic approximation is not strictly (explicitly) evolutionary. For arbitrary vertical structure, there is an approach to make it evolutional by vertical discretization to a set of shallow water systems, you can see the reference in my comments https://arxiv.org/abs/2001.08637 . With such an approach hyperbolicity can be strictly analyzed, so the conclusions on well-posedness were made, but it seems you just do not want to consider these works and to compare

to your results. Your approach with dependent variables and with 3 diagnostic and 3 prognostic equations seems questionable to me.

Ilias Sibgatullin

---

## Short Comment (SC21) · 12 Oct 2020

Dear Robert Nigmatulin and Xiulin Xu,

**1   "However, you are not satisfied with the process of estimating A. Thus, we only give an estimation of B"**

It's not about my satisfaction, but whether your estimation of A (the horizontal divergence of mass flux) in this manuscript and in

[Figure]

`https://doi.org/10.1134/S0015462818040201`, was correct for the considered scales. I've shown that it is wrong at synoptic scale, and I would expect a discussion from the point of view of logic and mathematics, rather than pleasing me.

Now I will show again that the estimation of B, the advection of pressure, given in `https://doi.org/10.1134/S0015462818040201` and replicated in your manuscript is also wrong for synoptic scales. You have estimated it as

$$\frac{\partial p}{\partial x}, \frac{\partial p}{\partial y} = \rho \, O \left( \frac{V_{hor}^2}{L_{hor}} + \frac{V_{hor}^2}{L_{cor}} \right).$$

For a general case of large-scale atmospheric dynamics this estimation is wrong for the reasons, I've already mentioned in my previous messages SC10 and comments on pubpeer.com.

Recall again the exact expression (SC9) for the total derivative of pressure in case of hydrostatic approximation:

$$\frac{dp}{dt} = \quad g \int_z [-div_{hor}(\rho \vec{v}_{hor})]dz + \quad \vec{v}_{hor} \nabla_{hor} p$$

I gave you a straightforward scale analysis of the pressure advection by directly substituting the expression of pressure as weight of the atmospheric column in the SC14 message and pubpeer.com, but you just ignore it.

$$\frac{dp}{dt} = \quad g \int_z [-div_{hor}(\rho \vec{v}_{hor})]dz + \quad g \vec{v}_{hor} \int_z \nabla_{hor} \rho dz$$

Now I repeat the counterexample to your estimation from SC10. At synoptic scales the pressure gradient force is almost balanced by Coriolis force. From the geostrophic balance

$$f \cdot v = \quad \frac{1}{\rho} \frac{\partial p}{\partial x} f \cdot u = -\frac{1}{\rho} \frac{\partial p}{\partial y},$$

the pressure gradient would rather be estimated as $\dfrac{\partial p}{\partial x} \approx \hat{\rho} f U$, than $\hat{\rho}\dfrac{U^2}{L}$. I wrote about that in SC10, but instead of answering about this estimation at synoptic scales, you have answered that Coriolis force is taken into account in the equations (it looks like an art of replying without answering questions). If you estimate $\dfrac{U}{fL}$ as $\dfrac{1}{10}$, which is common at synoptic scales, your estimation of ratio of the advection term to divergence term is wrong by at least two orders of magnitude.

Finally, both advection of pressure B and divergence of mass flux A, as well as total derivative of pressure, are small as compared to pressure in hydrostatic approximation.

**2   "The purpose of this paper is to analyze the shortwave instability of the quasi-hydrostatic equations."**

Is it really? Please look at your abstract:

> *An advanced "quasi-hydrostatic approximation" of 3-dimensional atmospheric-dynamics equations is proposed and justified with the practical goal to optimize atmospheric modelling at scales ranging from meso meteorology to global climate. For the vertically quasi-hydrostatic flow with inertial forces negligibly small compared to gravity forces, the asymptotically exact equation for vertical velocity is obtained...*

Since your terminology may differ from the conventional, is it so that you still stick to the horizontal scales from $\approx 1$ km for the system of equations given in this manuscript and in `https://doi.org/10.1134/S0015462818040201` ? Or you consider it wrong already and the scales of applications of your systems are now redefined?

[Figure]

The correctness of the instability analysis is another question, which is addressed in other comments.

**3    Please do not make any comments on anything that has nothing to do with the manuscript's content. Malicious comments on the authors are not welcomed.**

Dear Robert Nigmatulin and Xiulin Xu,

All my comments are about the manuscript `doi.org/10.5194/gmd-2020-146`. And since this manuscript up to (3.1) is a replication of another manuscript `doi.org/10.1134/S0015462818040201`, written by the First Author, I refer to that paper and critical reviews on that paper.

`doi.org/10.1134/S0015462818040201` was a subject of critical reviews over three years (even before its publication), and open comments to the Editorial Board of the journal where it was published in September 2019. The comments are published at arxiv.org and pubpeer.com since the beginning of 2020. This is why I refer to them also. Is it not true, or what is malicious about that?

The claims which are made in the Theorem are very important. The analysis of the total derivative of pressure, violation of the energy conservation, persistent non-citation of L.F. Richardson, strange notations, linearly dependant "independent" variables, strange results of linear analysis: all these issues are of minor importance, as compared to the major claim in your Theorem, reflected in the title of this manuscript

that hydrostatic approximation can be applied on account of the smallness of the vertical force of inertia,

Despite my persistent enquiries for a proof of such a statement for years, I see now the

Theorem with this claim being published again, and again without any proof.

I considered such a claim as a typical mistake of a student of the 3-d year, who have heard about small vertical velocity and small acceleration in models of shallow water equations or more generally in hydrostatic approximation, but had never bothered to look in the manuals for the reason of these approximations. And a very "logical" conclusion for such a student would be the conclusion of your Theorem: vertical acceleration is small, hence we get hydrostatic approximation. I'd just send such a student for reexaminations or explain the mistakes.

But the story here is not about a student or a researcher, who've published a mistake or overseen a previous research. Happily, if such a case could be forgotten. But here the paper `doi.org/10.1134/S0015462818040201` had received both private and public critical reviews. And after that the Theorem is again replicated here. The importance of this discussion is emphasised by the fact, that the First Author was recently appointed as the "scientific director" of the Institute of Oceanology in Moscow and head of the subdivision of Mechanics at Moscow University. Is it malicious or something wrong about mentioning this public information inhere? I explicitly remind about that inhere, to underline the fact, that the First Author have the power to influence on decisions about the fate of the scientific groups and teaching programs. In particular, the Theorem is incompatible with studies of internal waves. So the questions of mathematical correctness are particularly important here. (Is it not, or something is wrong or malicious?)

To my opinion, the questions here are quite serious to put it on the shoulders of a student who was not even an Author of the Theorem, which was formulated by the First Author earlier in `doi.org/10.1134/S0015462818040201`. I hope that the First Author will explicitly put his name under the replies (if any name will be present at all, I remember SC5 from the Executive Editor).

Best wishes, Ilias Sibgatullin.

---

## Short Comment (SC22) · 23 Oct 2020

Dear Robert Nigmatulin and Xiulin Xu,

A very detailed review on connection between correctness of Cauchy problem and hyperbolicity in the problems of Weather Prediction was given in the books by V. Gordin: Mathematical Problems of the Hydrodynamic Weather Forecast. Numerical Aspects. Mathematical Problems of the Hydrodynamic Weather Forecast. Analytical Aspects. Hydrometeorological Center of USSR, 1987

Also you can find the book V.A.Gordin. Mathematical Problems and Methods in Hydro-

dynamical Weather Forecasting. Gordon & Breach, 2000, 842p.

With respect to hydrostatic approximation the general idea is the following.

For evolution equations in partial derivatives of the first order correctness of the Cauchy problem is equivalent to hyperbolicity.

The system of equations in hydrostatic approximation is not explicitly evolutional, since there are equations without time derivative.

But is can be made evolutional after appropriate vertical discretizations. And it was shown analytically, that the resulting system is not necessarily hyperbolic, for unstable stratifications it is not.

For certain restrains on stratifications and vertical discretizations the hydrostatic approximation can be considered as a system of shallow water equations with different sound velocities (velocities of propagations of small perturbations). $\sqrt{gH}$ ("sound speed" in shallow water) corresponds here to $\sqrt{\lambda_i}$ where $i$ is the number of the layer. The system can lose hyperbolicity when $\lambda_i$ becomes negative. Sometimes small scale mixing or convective adjustment can resolve this problem.

―――――――――――――――――――――

---

## Author Comment (AC3) · 30 Oct 2020

To finalize the discussion part, we have the following statements regarding the questions that occurred in the discussion.

1. This work aims to study the shortwave stability property of the vertically quasi-hydrostatic system of equations using the linearized equations for perturbations. This system of equations is applicable and used for almost all large-scale climatic and meteorological calculations with characteristic time greater than $\tau \geq 10^2$ s, spatial scales over $L \geq 10^3$ m and velocity scale $V \sim L/\tau \leq 10$ m/s. Four

Interactive
comment

types of evaluation for vertical velocity (or pressure) are compared on the base of vertically quasi-hydrostatic approximation:

(a)     The asymptotically exact equation (fourth equation of (2.23))

(b)     Holton's approximation (4.23)

(c)     Quasi-incompressibility of air particle (4.28)

(d)     Quasi-incompressibility in space point (Marchuk's approximation) (4.30)

2. We note that Sibgatullin's criticism aims at the asymptotically exact equation (a), published in the paper of the first co-author in 2018, and it is not the subject of our manuscript. In the manuscript it is shown that in the equation for vertical velocity, the term with the substantial pressure derivative divided by pressure ($p^{-1}\mathrm{d}p/\mathrm{d}t$) consists of four components

$$\frac{1}{p}\frac{\mathrm{d}p}{\mathrm{d}t} = \frac{1}{p}\frac{\partial p}{\partial t} + \frac{1}{p}v_x\frac{\partial p}{\partial x} + \frac{1}{p}v_y\frac{\partial p}{\partial y} + \frac{1}{p}v_z\frac{\partial p}{\partial y}$$

In the paper 2018 it was shown that the terms, connected with horizontal relative pressure transfer (the second and the third components in the right side of the last expression) tend to zero asymptotically. We have also answered Ilias Sibgatullin many times, both at seminars in Russia and during the manuscript discussion. He does not want to understand our straightforward arguments repeated in this discussion. In particular, we have answered Sibgatullin on how to conduct the estimation

$$\frac{1}{p}v_y\frac{\partial p}{\partial y},\, \frac{1}{p}v_x\frac{\partial p}{\partial x} \sim \frac{\mathsf{M}^2}{\tau} \sim \frac{\varepsilon}{\tau} \to 0, \quad \mathsf{M} = \frac{V}{C} \leq 10^{-3}, \quad C = \frac{\gamma p}{\rho} \sim 300\,\mathrm{m/s}$$
$$(see\, SC16\, and\, SC8),$$

where $M$ and $C$ are Mach Number and sound speed. It means that horizontal relative pressure transfer asymptotically tends to zero for $M \sim \epsilon \to 0$. Instead of this estimation, Sibgatulin permanently considers the horizontal pressure transfer terms (without $1/p$)

$$v_x \frac{\partial p}{\partial x}, \; v_y \frac{\partial p}{\partial y} \quad (see\, SC7).$$

Therefore, further discussion on this issue makes no sense. After all, equation (a) (or the fourth equation of (2.23)) may be considered as one of the approximations.

3. It is shown that the shortwave perturbations with a large ratio of horizontal wavenumbers to the vertical wavenumber will make the solutions (even the resting state solutions) of quasi-hydrostatic systems (A, B, C, D) unstable. As shortwave perturbations occur during the numerical calculation of the original differential equations, and the wavelengths of the perturbations are proportional to the grid sizes of meshing, the result of shortwave stability can be used for appropriate meshing to achieve stable numerical calculation.

4. The dispersion relation for gravity waves is different from that in our result (from anonymous referee #2, see RC2). We state all the differences between our set of equations and the equations used in RC2 in the reply SC13. The perturbations in RC2 are assumed to be in harmonic form with constant amplitude, while we allow the amplitude of perturbations to change by time, as in the manuscript (3.11).

5. More doubts about the quasi-hydrostatic equations' applicability occur from anonymous referee #1 (see RC1). As described in the manuscript, the quasi-hydrostatic approximation is only valid for long-term climatic processes (when all inertia forces are negligibly small compared with the gravity force). It is important to realize that the neglect of inertial forces appears only in the equation of vertical velocity. And it makes the system of equations (2.23) non-hyperbolic with

an infinite speed of sound, then its solution is unstable to harmonic shortwave perturbations when the solution evolves forward in time. Taking into account the small vertical inertia force makes the numerical meteorological and climatic calculations non-realistic because of extremely small time steps. The energy of the system is conserved as the thermodynamic equation is used to evaluate the vertical velocity. However, the perturbations' energy is not constrained because they are caused by numerical error and filled in the entire space. A detailed response to anonymous referee #1 is in AC2.

We are trying to improve the manuscript to make it clear with points 3, 4 and 5 marked by anonymous referees # 1 and # 2. We are very grateful to them for their remarks.

Sincerely,

Robert Nigmatulin and Xiulin Xu